# Integrator complex subunit 12 knockout overcomes a transcriptional block to HIV latency reversal

Carley N Gray[1], Manickam Ashokkumar[2,3], Derek H Janssens[4†],
Jennifer L Kirchherr[3], Brigitte Allard[3], Emily Hsieh[5], Terry L Hafer[4,6],
Nancie M Archin[2,3], Edward P Browne[2,3,7], Michael Emerman[4,6]*

[1]Department of Microbiology, University of Washington, Seattle, United States;
[2]Division of Infectious Diseases, Department of Medicine, University of North Carolina at Chapel Hill, Chapel Hill, United States; [3]UNC HIV Cure Center, University of North Carolina at Chapel Hill, Chapel Hill, United States; [4]Division of Basic Sciences, Fred Hutchinson Cancer Center, Seattle, United States; [5]Molecular and Cellular Biology Graduate Program, University of Washington, Seattle, United States; [6]Division of Human Biology, Fred Hutchinson Cancer Center, Seattle, United States; [7]Department of Microbiology and Immunology, University of North Carolina at Chapel Hill, Chapel Hill, United States

*For correspondence:
memerman@fredhutch.org

Present address: †Department of Epigenetics, Van Andel Institute, Grand Rapid, United States

Competing interest: The authors declare that no competing interests exist.

## eLife Assessment

Using multiple techniques previously validated by the authors, this study identified INTS12, a component of the Integrator complex involved in 3' processing of small nuclear RNAs U1 and U2, as a factor promoting HIV-1 latency. The work is **valuable**, based on a sound strategy for screening targets to activate HIV latency and the **solid** mechanistic insights it provides on INTS12 repression of transcriptional elongation. Future studies are needed to explore INTS12 as a drug target against HIV/AIDS.

**Abstract** The latent HIV reservoir is a major barrier to HIV cure. Combining latency reversal agents (LRAs) with differing mechanisms of action such as AZD5582, a non-canonical NF-kB activator, and I-BET151, a bromodomain inhibitor is appealing toward inducing HIV-1 reactivation. However, even this LRA combination needs improvement as it is inefficient at activating proviruses in cells of people living with HIV (PLWH). We performed a CRISPR screen in conjunction with AZD5582 & I-BET151 and identified a member of the Integrator complex as a target to improve this LRA combination, specifically Integrator complex subunit 12 (INTS12). Integrator functions as a genome-wide attenuator of transcription that acts on elongation through its RNA cleavage and phosphatase modules. Knockout of INTS12 improved latency reactivation at the transcriptional level and is more specific to the HIV-1 provirus than AZD5582 & I-BET151 treatment alone. We found that INTS12 is present on chromatin at the promoter of HIV and therefore its effect on HIV may be direct. Additionally, we observed more RNAPII in the gene body of HIV only with the combination of INTS12 knockout with AZD5582 & I-BET151, indicating that INTS12 induces a transcriptional elongation block to viral reactivation. Moreover, knockout of INTS12 increased HIV-1 reactivation in CD4 T cells from virally suppressed PLWH ex vivo, and we detected viral RNA in the supernatant from CD4 T cells of all three virally suppressed PLWH tested upon INTS12 knockout, suggesting that INTS12 prevents full-length HIV RNA production in primary T cells. Finally, we found that INTS12 more generally limits the efficacy of a variety of LRAs with different mechanisms of action.

## Introduction

Despite advancements in keeping viral loads below detectable limits, HIV-1 still exists within a latent reservoir in vivo and viral replication returns when antiretroviral therapy (ART) is interrupted (*Davey et al., 1999*). HIV-1 predominately infects memory CD4+ T cells (reviewed in *Ait-Ammar et al., 2019*; *Cohn et al., 2020*), and while many acutely infected cells, particularly the more differentiated effector memory cells, die off rapidly upon infection due to cytopathic effects, some cells can revert to a resting state where the provirus becomes transcriptionally silent (reviewed in *Duggan et al., 2023*; *Mbonye and Karn, 2024*). These infected resting memory cells form the HIV reservoir that is largely invisible to the immune system and not sensitive to enhanced ART (*Siliciano and Siliciano, 2024*). The reservoir of latently infected cells decays slowly and is maintained by both external stimuli and homeostatic proliferation even after initiation of ART (reviewed in *Cohn et al., 2020*). Thus, the major barrier to HIV cure is the existence of this ART-insensitive and long-lived latent reservoir that is largely not sensed by the immune system (*Chun and Fauci, 1999*), reviewed in *Duggan et al., 2023*; *Mbonye and Karn, 2024*.

One strategy being explored to target the latent reservoir, called 'shock-and-kill', uses small-molecule compounds called latency reversal agents (LRAs) to transcriptionally activate latent cells, so that immune-mediated mechanisms can then recognize and kill infected cells (*Archin et al., 2012*), reviewed in *Maina et al., 2021*. This strategy would be employed in the presence of ART so that newly induced HIV virions could not infect more cells. However, inducing HIV-1 activation from latency is challenging because the latent reservoir size varies between people and heterogeneous HIV integration sites can influence the reactivation of infected cells (reviewed in *Ait-Ammar et al., 2019*). Moreover, activation of latently infected resting T cells has not yet been able to induce HIV-1 replication from more than a fraction of the cells with a replication-competent provirus (*Ho et al., 2013*; *Siliciano and Siliciano, 2024*).

Transcriptional repression of HIV-1 in its latent state is multifactorial, and it is known to be affected by repressive chromatin modifications and blocks to both transcription initiation and transcription elongation. Several classes of LRAs have been identified that can undo one or more blocks to latency reversal (*Stoszko et al., 2019*), reviewed in *Rodari et al., 2021*; *Singh et al., 2021*. For example, AZD5582 is an LRA that targets a transcription initiation block by activating the noncanonical NF-kB pathway (*Hennessy et al., 2013*; *Pache et al., 2020*) and has been shown to induce HIV transcription in HIV latency cell line models as well as induce viremia in animal models (*Nixon et al., 2020*). Another class of LRAs are BET inhibitors that undo a block to transcription elongation by targeting various bromodomain-containing proteins. I-BET151 is a pan-BET inhibitor that is thought to reverse HIV latency by either blocking host Brd4 from binding to P-TEFb to facilitate P-TEFb availability for HIV Tat (*Jang et al., 2005*; *Schröder et al., 2012*; *Yang et al., 2005*) or more generally by increasing the amount of P-TEFb not bound to the 7SK complex (*Turner et al., 2024*). Currently, no single LRA has been potent enough to reactivate the majority of the latent proviruses nor specific enough to HIV to not cause cytotoxicity (reviewed in *Debrabander et al., 2023*). On the other hand, combinations of LRAs with different mechanisms of action have shown synergy in HIV latency reversal in different model systems including the combination of AZD5582 & I-BET151 (*Dai et al., 2022*; *Falcinelli et al., 2022*; *Laird et al., 2015*) reviewed in *Rodari et al., 2021*. However, while the combination of AZD5582 & I-BET151 is synergistic in activation of HIV in latency cell line models, and can induce robust HIV transcription initiation ex vivo in primary CD4+ T cells from virally suppressed people living with HIV (PLWH), it failed to induce full-length proviral transcription in the ex vivo system (*Falcinelli et al., 2022*). Thus, there exist additional blocks to HIV reactivation that even potent LRA combinations currently cannot overcome.

Here we hypothesized that we could use a CRISPR screen to uncover pathways to improve LRA combinations. Previous work from our lab used a high-throughput CRISPR screening approach, called the Latency HIV-CRISPR screen, to predict gene knockouts that improve the efficacy of a single LRA, AZD5582. Specifically, knockout of ING3, a member of the NuA4 histone acetylation complex, could improve the potency and the specificity of AZD5582, both in J-Lat model systems and in a primary T cell model of latency. Thus, in this study, we used a similar strategy to identify gene knockouts to increase HIV latency reversal with the more powerful AZD5582 & I-BET151 drug combination, and uncovered Integrator complex subunit 12, INTS12, as contributing to a block in HIV reactivation, both on its own, but, especially in the presence of the AZD5582 & I-BET151 drug combination.

INTS12 is a subunit of the Integrator complex that is thought to act as a reader to link the complex to chromatin (*Welsh and Gardini, 2023*). The Integrator complex functions as a genome-wide attenuator of mRNA transcription where it can antagonize transcription with both its cleavage and phosphatase modules (*Hu et al., 2023*), and members of Integrator have been shown to modulate transcription at the HIV LTR previously (*Lykke-Andersen et al., 2021*; *Stadelmayer et al., 2014*).

We found that knocking out INTS12 reactivates HIV on its own and improves reactivation with AZD5582 & I-BET151 treatment. RNA-seq analysis showed that INTS12 knockout (KO) more specifically reactivates HIV compared to AZD5582 & I-BET151 treatment alone, and the combination of INTS12 KO with AZD5582 & I-BET151 increases the potency of reactivation. Additionally, we have evidence of an elongation block being overcome with INTS12 KO paired with AZD5582 & I-BET151 specifically showing an increase in bound elongation-competent RNAPII in the gene body of HIV that is not seen with either KO or LRA treatment alone. We found that INTS12 is bound to the viral promoter and therefore may be acting directly on viral transcription. Importantly, we saw reactivation of HIV in the CD4 T cells of the majority of the virally suppressed PLWH tested ex vivo, upon INTS12 KO with or without AZD5582 & I-BET151 treatment. Crucially, we also detected viral RNA in the supernatant suggestive of the presence of HIV virions released into the supernatant, indicating that full-length HIV transcripts are being generated. Furthermore, we used the HIV-CRISPR screen approach to probe the Integrator complex as a whole and predict Integrator members comprising the cleavage module, phosphatase module, and core to all be involved in the control of HIV-1 latency in addition to the reader, INTS12. Lastly, we show that knockout of INTS12 can increase reactivation with diverse classes of LRAs, suggesting that INTS12 is a general block to HIV reactivation.

## Results

### Latency HIV-CRISPR screen identifies gene targets that increase HIV latency reversal with AZD5582 & I-BET151 treatment

We used a high-throughput CRISPR screen previously described by our lab (*Hsieh et al., 2023*) to identify gene targets that, when knocked out, would further improve LRAs. The screen involves a lentiviral vector, called HIV-CRISPR, that contains two functional LTRs, a packaging signal, a guide RNA, and Cas9 to knock out a gene of interest (*Figure 1A*). Cells that have a latent HIV provirus are transduced with a library of HIV-CRISPR-containing guides and then, after selection for integration of the vector, the cells are treated with the combination of LRAs, AZD5582 & I-BET151 to reactivate HIV (*Figure 1A*). If a gene knockout improves LRA reactivation, we will detect more HIV in the supernatant compared to the DMSO control. We can determine the gene knockout responsible for HIV reactivation because the HIV-CRISPR vector will be packaged along with the reactivated provirus from the cells and become enriched in the supernatant (*Figure 1A*, see purple or black HIV RNA in virions). We then determine how enriched a guide is by the ratio of guides in the supernatant compared to the total guide representation in the cells, and this generates a value called the MAGeCK gene score, where higher values correspond to more enrichment (i.e., HIV reactivation) (*Li et al., 2014*). Importantly, all guides to one gene target and replicates are factored into the MAGeCK gene score generation.

We used a custom human epigenome guide library (HuEpi library) that contains 5309 sgRNAs total (839 genes with six guides/gene) that comprises histone binders/modifiers and other general chromatin-associated proteins, which casts a wide net on general epigenetic factors (further description of this library and the major gene ontology categories targeted has been published; *Hsieh et al., 2023*). We previously validated the HuEpi library as containing guides to multiple genes involved in HIV-1 latency (*Hsieh et al., 2023*). Here, we conducted additional screens in two cell lines, J-Lat 10.6 and J-Lat 5A8, that are clonal Jurkat T-lymphocyte cell lines that contain latent HIV. We chose these cells because they differ in integration site and HIV drug reactivation profiles (*Chan et al., 2013*; *Jordan et al., 2003*; *Spina et al., 2013*; *Telwatte et al., 2019*) and wanted to identify commonly enriched genes that were not integration site specific.

The screen results are shown as a comparison of a DMSO-treated screen compared to a screen carried out in the presence of 10 nM AZD5582 & 100 nM I-BET151 combined (*Figure 1B and C*). Candidate knockouts that are predicted to improve the AZD5582 & I-BET151 combination specifically should fall above the dotted line in *Figure 1B and C* as these genes are more highly enriched in the AZD5582 & I-BET151 screen compared to the DMSO screen. Genes that fall on the line are predicted

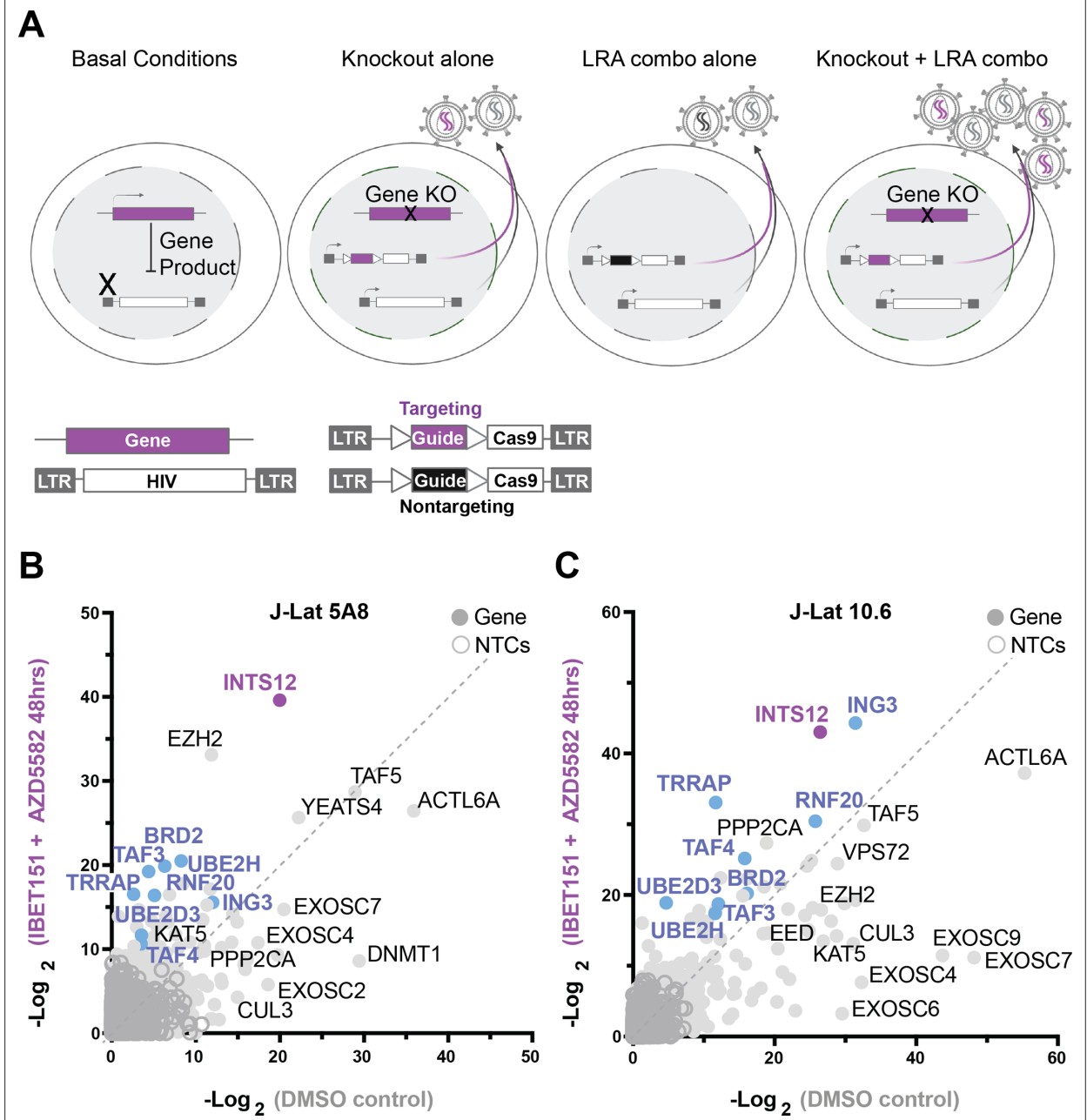

**Figure 1.** A screen to predict gene knockouts that will improve HIV reactivation from the combination of AZD5582 & I-BET151. (**A**) Basal J-Lat conditions (left panel) followed by screen scenarios (middle to right panels). Screen overview: J-Lat cells containing an internal provirus are transduced with the HIV-CRISPR vector to generate a library of knockout cells for the human epigenetic library. Knockout cells are selected by puromycin and then either treated with latency reversal agents (LRAs) or untreated (DMSO). The HIV-CRISPR vector is packageable and will accumulate in the supernatant if the internal HIV provirus is reactivated. Sequencing the supernatant and cells allows for measurement of how enriched a guide is. (**A**, from left to right) Basal conditions show a gene product can block reactivation of HIV at the transcriptional level. Knockout alone shows that upon introducing the HIV-CRISPR some gene knockouts will result in latency reversal in the absence of LRA treatment. LRA combo alone shows the effect of LRA stimulation alone and how this will result in non-targeting guides (black) accumulating in the supernatant. Knockout + LRA combo shows that some gene knockouts will improve reactivation with LRA treatment and result in more virus accumulation. (**B, C**) are the results of the screens in J-Lat 5A8 (**B**) and J-Lat 10.6 (**C**). Each is graphed as a comparison of the LRA combination AZD5582 & I-BET151 (Y-axis) treated screens to the untreated (DMSO) screens (X-axis) as the -Log$_2$ MAGeCK gene scores for each gene target. Purple and blue genes are genes in common between J-Lat 10.6 and 5A8 that are predicted to improve AZD5582 & I-BET151 treatment specifically, and purple is the top gene hit in common. NTC = nontargeting control guide.

The online version of this article includes the following figure supplement(s) for figure 1:

**Figure supplement 1.** Validation of HIV-CRISPR screen with inhibitors.

to be drug-independent hits as they are equally enriched in both screens, and genes falling below the line are thought to be redundant to the function of the LRAs.

With this approach, we identified gene hits that have been validated previously using the same HuEpi library (*Hsieh et al., 2023*): KAT5, CUL3, ACTL6A, TAF5, DNMT1, DMAP1, VPS72, YEATS4, SRCAP, and ING3 (*Figure 1B and C*, full screen list and results in *Supplementary file 1*). Knockout of ING3 was previously shown to improve AZD5582 treatment for activation of latent HIV-1 (*Hsieh et al., 2023*), and also shows up as a candidate hit in the present screen done the presence of both AZD5582 & I-BET151 (*Figure 1B and C*). Our top hit, INTS12, here was also seen in our previous study (*Hsieh et al., 2023*), but was not followed up on in that study.

Gene hits of interest that we identified that are predicted to improve the AZD5582 & I-BET151 combination specifically in both J-Lat lines tested including (shown in blue or purple, *Figure 1B and C*): INTS12, TRRAP, RNF20, BRD2, TAF3, TAF4, UBE2D3, and UBE2H. Some hits, EZH2 and PPP2CA, were only predicted to improve AZD5582 & I-BET151 treatment in one cell line. Because EZH2 has a number of available inhibitors, we validated the screen with the EZH2 inhibitor Tazemetostat (*Straining and Eighmy, 2022*). That is, the screen predicted that inhibition of EZH2 would increase the potency of AZD5582 & I-BET151 to activate HIV-1 from latency in J-Lat 5A8 cells. Indeed, we found that Tazemetostat increases reactivation of HIV-1 in the presence of AZD5582 & I-BET151 to much higher levels than AZD5582 & I-BET151 alone or Tazemetostat alone (*Figure 1—figure supplement 1*). However, we focused the remainder of this study on the most top-ranking hit common to both J-Lat models that was enriched in the combination AZD5582 & I-BET151 treatment, Integrator complex subunit 12 (INTS12).

## Integrator complex subunit 12 knockout specifically reactivates HIV independently and in combination with AZD5582 & I-BET151 treatment

The Integrator complex is known to play a role in the negative regulation of elongating transcripts (*Elrod et al., 2019*; *Tatomer et al., 2019*). Moreover, previous studies have shown that some Integrator members are recruited to the HIV LTR and regulate HIV transcript length (*Stadelmayer et al., 2014*), although INTS12 had not been previously implicated in HIV regulation. We validated INTS12 as a screen hit by either knocking out INTS12 with the HIV-CRISPR vector loaded with one guide and then selecting for knockout cells with puromycin (J-Lat 5A8 [*Figure 2A*], J-Lat 10.6 [*Figure 2—figure supplement 1*]) or by electroporation with a combination of three of guides to INTS12 (*Figure 2B and C*). We used AAVS1 knockout as a CRISPR control where the guide targets within a safe harbor locus and knockout should have a minimal effect on transcription (*Li et al., 2014*). These knockouts were then treated with DMSO, a low dose of AZD5582 (10 nM), a low dose of I-BET151 (100 nM), or a combination of AZD5582 (10 nM) & I-BET151 (100 nM) (*Figure 2A and B*), and the amount of HIV-1 released into the supernatant was measured by a reverse transcriptase activity assay (*Vermeire et al., 2012*). As expected, we find in AAVS1 knockout cells that there is minimal reactivation in the DMSO control, a small amount of HIV induction from AZD5582 or I-BET151 alone, and we observe synergy with AZD5582 & I-BET151 used in combination (*Figure 2A and B*). Knocking out INTS12 alone in the absence of added LRAs also shows a modest increase in the amount of virus detected in the supernatant (*Figure 2A and B*, DMSO), indicating that INTS12 itself is imposing a block to reactivation that is relieved upon knockout. In the presence of drug treatments, INTS12 knockout improves reactivation of either AZD5582 or I-BET151 alone, and we see the most HIV reactivation when INTS12 knockout is treated with AZD5582 & I-BET151 in combination (*Figure 2A and B*), as the screen predicted (*Figure 1B and C*). Flow cytometry data using the expression of GFP in the provirus gave similar results and showed a greater percentage of cells are activated from INTS12 KO alone and in combination with AZD5582 & I-BET151 (*Figure 2—figure supplement 2A and B*). To ensure cell viability did not contribute to this phenotype, we measured cell proliferation in our AAVS1 KO and INTS12 KO cells over time and observed no differences in cell growth (*Figure 2—figure supplement 3*).

Furthermore, to ensure that this effect was due to knockout of INTS12 and not an off-target CRISPR effect, we performed INTS12 complementation experiments. We took pooled INTS12 KO cells and AAVS1 KO cells and transduced them with a lentiviral vector containing INTS12 and a puromycin resistance gene, selected for vector transduction, and then measured HIV reactivation (*Figure 2C*). We found that reactivation of the HIV-1 provirus in the INTS12 KO cells that were complemented with an INTS12 expression vector were now equivalent to the control AAVS1 KO cells with the same INTS12

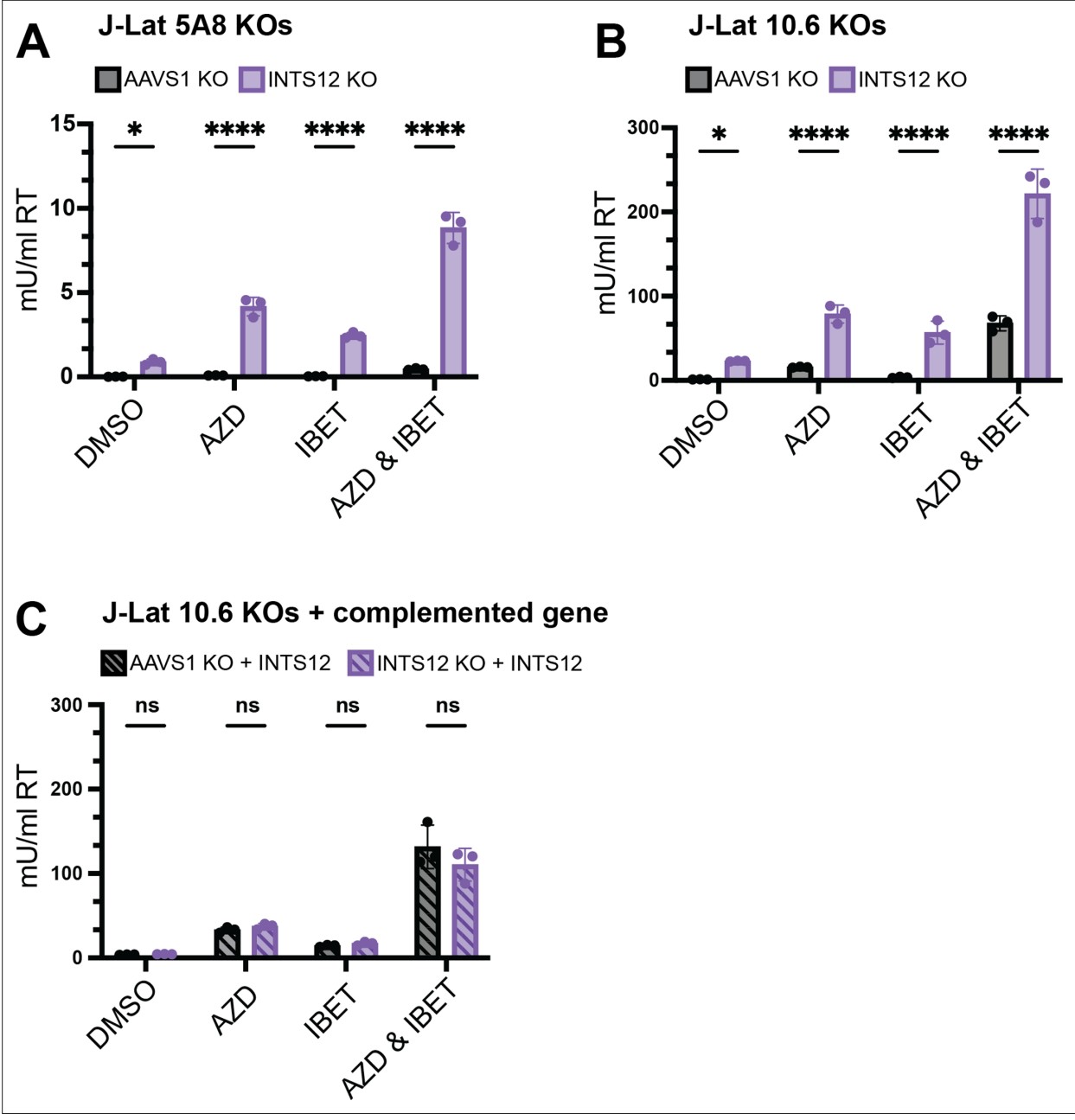

**Figure 2.** Validation of INTS12 knockout in HIV latency reversal both on its own and in the presence of AZD5582 & I-BET151. J-Lat 5A8 (**A**) and J-Lat 10.6 (**B**) cells were knocked out for INTS12 or control locus, AAVS1. With a calculated INTS12 knockout score of 76% (for the one guide used) and 69% (for one of three guides used), respectively. Cells were then treated with 10 nM AZD5582 and or 100 nM I-BET151 for 48 hr (or an equivalent volume of DMSO), and HIV reverse transcriptase activity was measured from the supernatant (reported in mU/mL). (**C**) Complementation of cells knocked out for INTS12 or AAVS1 was transduced with a vector containing INTS12 before latency reversal agent (LRA) treatments. These cells had an INTS12 knockout score of 55% for one of three guides used. Untreated = DMSO, AZD = AZD5582, IBET = I-BET151. For statistical analysis, all conditions are compared to the AAVS1 control. n=3 replicates for each condition. Two-way ANOVA, (**A, B**) uncorrected Fisher's LSD (**C**), Šídák's multiple comparisons test, *p-value<0.05, **<0.01, ***<0.001, ****<0.0001.

The online version of this article includes the following figure supplement(s) for figure 2:

**Figure supplement 1.** Validation of INTS12 knockout in HIV latency reversal both on its own and in the presence of AZD5582 & I-BET151.

**Figure supplement 2.** Validation of INTS12 knockout in HIV latency reversal both on its own and in the presence of AZD5582 & I-BET151.

**Figure supplement 3.** Growth curve of INTS12 knockout cells.

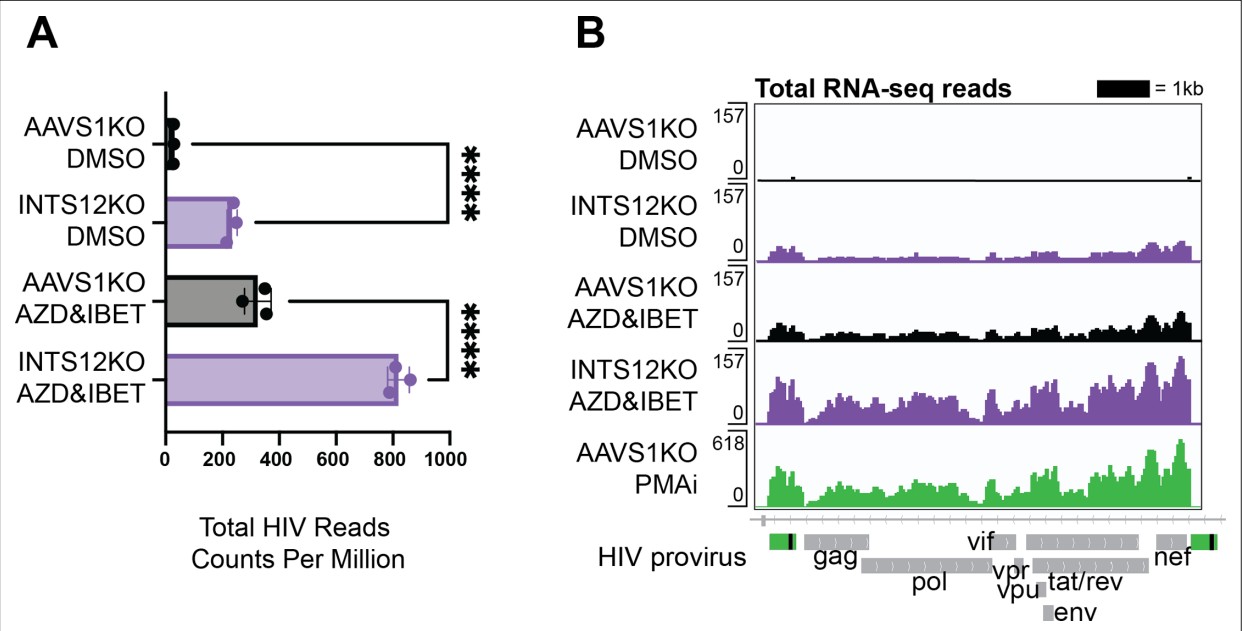

**Figure 3.** INTS12 KO reactivates HIV alone and in combination with AZD5582 & I-BET151 at the level of steady-state viral transcripts. (**A**) Total RNA-seq reads mapping to the HIV provirus were averaged for biological triplicates normalized to copies per million reads (CPM). One way ANOVA, Šídák's multiple comparisons tests, *p-value<0.05, **<0.01, ***<0.001, ****<0.0001. Read counts of the HIV-1 provirus in the AAVS1 KO treated with DMSO average about 30 CPM while read counts of the HIV-1 provirus in the INTS12 KO treated with AZD5582 and IBET-151 average about 800 CPM. (**B**) Pileup graphs corresponding to each of the conditions represent averaged reads from biological triplicates that have been normalized to CPM, and the peaks correspond in location to the integrated provirus with viral genes labeled below. The scale of the pileups is on the top left of each row. Note that PMAi has a different scale than the other samples. The 3' LTR is masked so that LTR-containing reads will only map to the 5' LTR. AZD = AZD5582, IBET = I-BET151, PMAi = PMA and ionomycin. All chromosome locations and quantified regions can be found in the 'Materials and methods'. The INTS12 cell pools used in this figure have an INTS12 knockout score of 75%.

expression vector (*Figure 2C*). These results show that INTS12 is responsible for a negative effect on HIV expression in the presence of LRAs, AZD5582 & I-BET151.

## INTS12 knockout increases HIV steady-state RNA levels with specificity for the LTR

To test the effects of INTS12 knockout on both HIV-1 and total host transcription, we performed RNA-seq on total RNA that had been depleted for ribosomal RNA. RNA-seq was done on pooled INTS12 knockouts and pooled AAVS1 knockouts (as controls) in J-Lat 10.6 cells in biological triplicates, where knockout of cells was generated three separate times and treated with either DMSO or AZD5582 & I-BET151. We first looked at how the sum of all transcripts mapping to HIV changed with the different conditions (*Figure 3A*). We observed that the RNA-seq trend matches what we saw with the virus release assay (*Figure 2B*) where the INTS12 KO alone reactivates HIV, and INTS12 KO further improves reactivation with AZD5582 & I-BET151. Reads were also mapped along the length of the HIV provirus (*Figure 3B*) using PMA and ionomycin (PMAi) -treated cells in this comparison as a positive control known to induce full-length transcripts (*Falcinelli et al., 2022*; *Figure 3B*, bottom row). We observe the same differences quantified in (*Figure 3A*) as well as similar pileup trends along the length of the provirus. Because RNAs lacking a polyA tail are degraded rapidly (*Biziaev et al., 2024*), it is unclear whether those RNAs were successfully captured in our assay, and it is likely we are only looking at full-length RNA detection. Nevertheless, the total RNA-seq results show that INTS12 KO increases HIV proviral activation at the transcriptional level.

We also investigated how INTS12 KO and AZD5582 & I-BET151 treatments alone and in combination affect transcription in the cell at a global level to assess the specificity to HIV compared to host genes. We first compared our AZD5582 & I-BET151 treatment condition to the DMSO-treated AAVS1 KO cells and found that 80 genes were upregulated and 306 were downregulated, and that HIV was the most upregulated transcript with the highest fold-change (FC) and was highly expressed

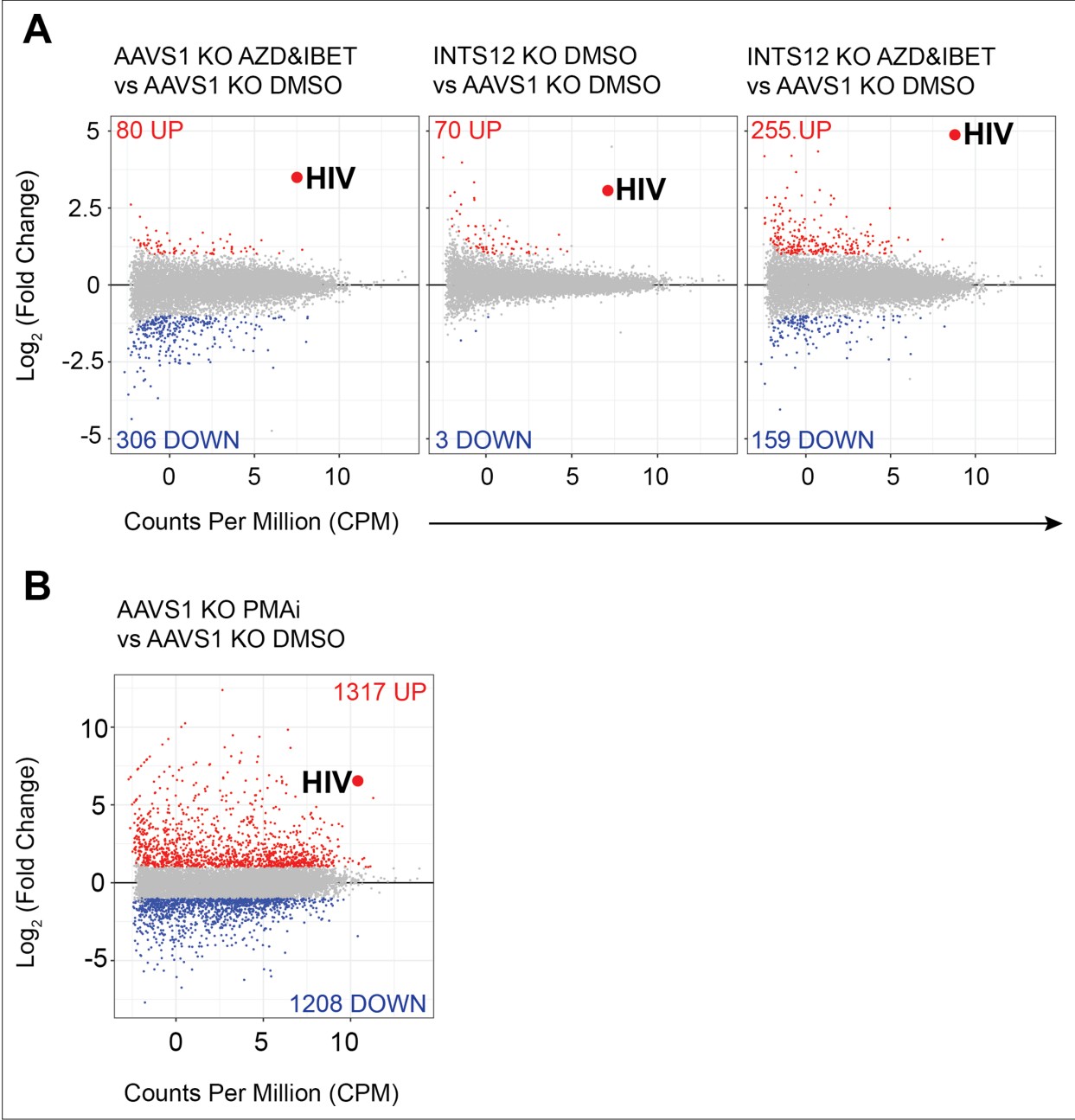

**Figure 4.** Specificity of INTS12 KO for reactivation of HIV on its own and in combination with AZD5582 & I-BET151. Differentially expressed genes are graphed by Log$_2$ (fold change) on the y-axis, denoting how upregulated (red) or downregulated (blue), a gene is, and the average counts per million (CPM) on the x-axis, denoting the expression of each gene. The big red dot denotes the average of all reads mapping to the HIV provirus. The red number at the top of each graph corresponds to the number of significantly upregulated genes in a comparison and the blue number at the bottom corresponds to number of significantly downregulated genes in a comparison. Gray genes are not statistically significant and not up or downregulated. (**A** left panel) shows the effect of AZD5582 & I-BET151 treatment alone compared to the DMSO control. (**A**, middle panel) shows the effect of the INTS12 KO on gene expression compared to the AAVS1 KO control. (**A**, right panel) shows the effect of both AZD5582 & I-BET151 treatment of INTS12 KO cells compared to the control DMSO-treated AAVS1 KO cells. (**B**) is graphed using a different scale than (**A**) and shows the effect of PMAi treatment compared to the DMSO control. The INTS12 cell pools used in this figure have an INTS12 knockout score of 75%.

as indicated by counts per million (CPM) (**Figure 4A**, left panel), similar to previous work (**Falcinelli et al., 2022**). Surprisingly, given the more widespread reported role of Integrator in transcription, we find that INTS12 KO alone is even more specific to HIV transcripts than AZD5582 & I-BET151 treatment, where we find the same level of enrichment of HIV as the previous comparison with an FC of 8, but there are only 70 upregulated genes and only 3 downregulated genes (**Figure 4A**, middle panel).

This indicates that INTS12 is more important to HIV transcription than it is to any other transcriptional unit in J-Lat cells.

Furthermore, the transcriptional landscape of the combination of INTS12 KO cells treated with AZD5582 & I-BET151 compared to AAVS1 KO cells that were treated with DMSO (*Figure 4A*, right panel) showed an even greater enrichment of the HIV transcripts at a FC of ~28 (Log$_2$(fold change) = 4.8). However, this gain of potency came with some loss of specificity as there was greater dysregulation of the host with 255 upregulated genes and 159 downregulated genes compared to the INTS12 knockout alone or the AZD5582 & I-BET151 treatment alone.

Among the host genes most prominently affected by INTS12 knockout with AZD5582 & I-BET151 are MAFA, MAFB, and ID2 (full list of genes and results in *Supplementary file 3*, tabs 1–4). Nonetheless, INTS12 knockout with AZD5582 & I-BET151 was much more specific to HIV-1 than PMAi, which can robustly activate HIV but at the cost of affecting thousands of host transcripts (*Figure 4B*). Thus, we find that INTS12 knockout acts at the transcriptional level in activating HIV from latency and is more specific for the HIV LTR than AZD5582 & I-BET151 treatment alone. We also find that INTS12 knockout indeed improves potency of reactivation of AZD5582 & I-BET151 but at the cost of some specificity.

## Chromatin localization reveals INTS12 may act directly on HIV transcription and INTS12 knockout paired with AZD5582 & I-BET151 overcomes an elongation block to HIV transcription

The effect of INTS12 on HIV-1 transcription could be direct (i.e., the Integrator complex directly affecting the steady-state levels of HIV-1 RNA transcription) or could be indirect by affecting the transcription of other gene products that subsequently act on the HIV-1 LTR. In order to partially distinguish these two hypotheses, we asked if INTS12 is present at the HIV-1 LTR using Cleavage Under Targets and Tagmentation (CUT&Tag) for profiling chromatin components (*Janssens et al., 2021*) using an antibody to INTS12 (*Figure 5A*). Using this approach, we observed that INTS12 is indeed at the LTR of HIV-1 as the signal over the LTR is greater in the AAVS1 KO cells than in cells with the partial INTS12 knockout (*Figure 5A*, left panel, compare top two rows with bottom two rows). We also saw this effect with a different INTS12 antibody (*Figure 5—figure supplement 1*). We found that that INTS12 localization extends past the LTR, matching what has been seen previously, where Integrator members localize to sites of pausing which are past the site of initiation (*Elrod et al., 2019*; *Stadelmayer et al., 2014*). In the right panel of *Figure 5A*, we quantify all reads in the region specified by the dotted gray lines in the left panel (*Figure 5A*, right panel; exact coordinates are in the 'Materials and methods'). These results indicate INTS12 binds to the HIV promoter, suggesting that INTS12 represses HIV-1 activation from latency through a direct mechanism.

Given the Integrator complex's role in pausing and termination at promoters, we wanted to look at HIV-1 elongation by measuring RNAPII levels as well as the Ser2 and Ser5 phosphorylation levels of the C-terminal tail of RNAPII with CUT&Tag. We first used an antibody against total RNAPII, RPB3, and generated pileup graphs of reads corresponding to where RPB3 is bound across the HIV-1 provirus and downstream (*Figure 5B*, left panel). The peaks in each row correspond to at least 3–5 technical replicates averaged together. We quantified all reads of RPB3 in three regions (*Figure 5B*, right panel; exact coordinates in the 'Materials and methods'). We see similar levels of RPB3 in region 1 for all four conditions. We defined region 2 as the remainder of the provirus excluding the LTR and the short 5' region of HIV in region 1. We found that there was a dramatic statistically significant increase in RPB3/RNAPII occupancy in the body of the provirus (region 2) only with both INTS12 KO and AZD5582 & I-BET151 together (*Figure 5B*). We did not see this increase in RNAPII when we examined the gene downstream of the integrated provirus (region 3). More RNAPII within the body of the HIV provirus is suggestive of transcription elongation, but to strengthen this claim we measured Ser2 phosphorylation of the C-terminal tail of RNAPII that is found on the elongation-competent form of RNAPII. We observed evidence of elongation-competent RNAPII for two conditions with LRAs: AAVS1 KO cells treated with AZD5582 & I-BET151 and our INTS12 KO cells treated with AZD5582 & I-BET151 (*Figure 5C*), as we saw a significant increase in reads in region 2 compared to the AAVS1 KO treated with DMSO control. When we tested RNAPII Ser5 phosphorylation, a marker not associated with elongation, we did not observe any statistically significant differences in any region for the same conditions tested in *Figure 5B and C* (*Figure 5—figure supplement 2*). Therefore, these data support

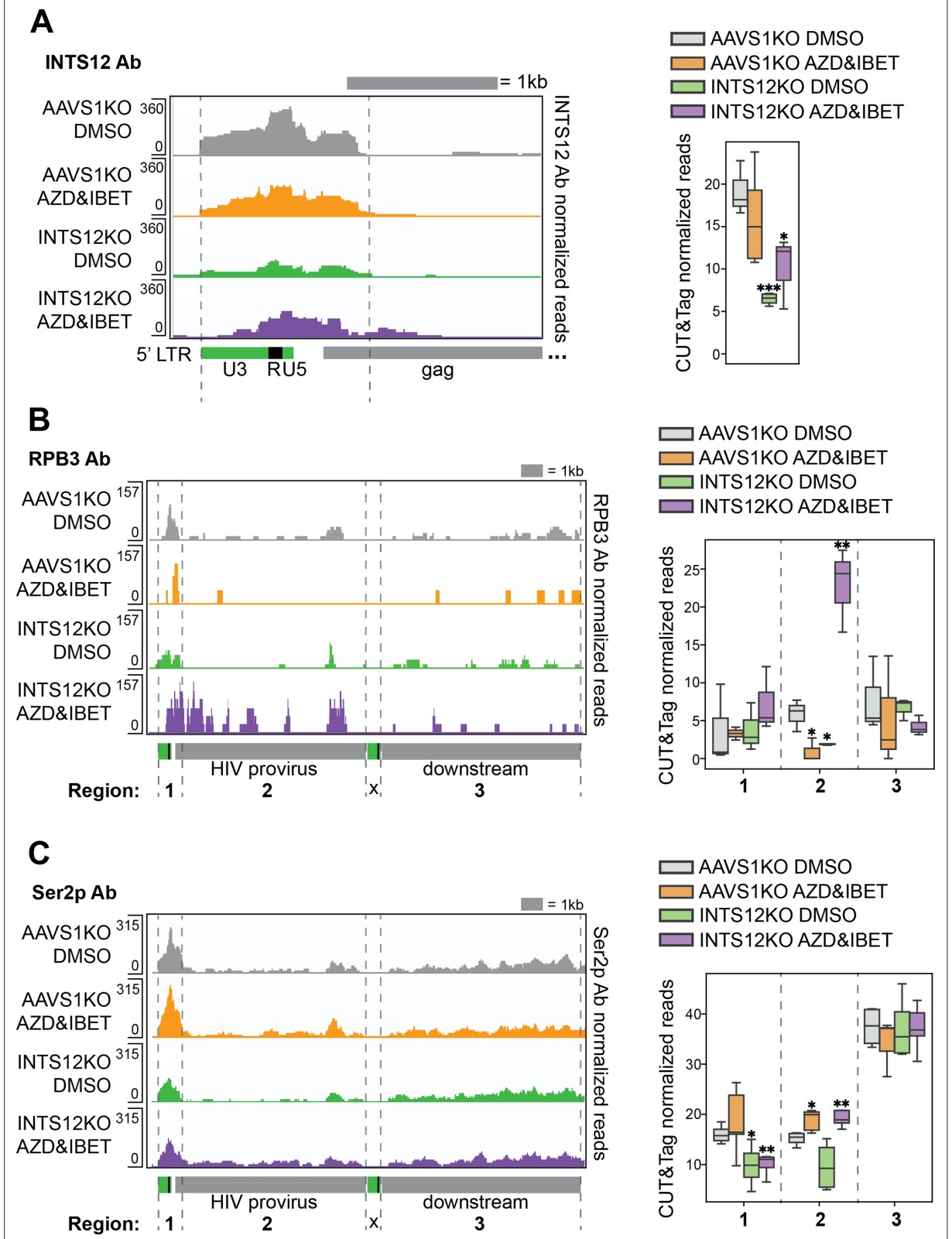

**Figure 5.** INTS12 binds the HIV promoter, and INTS12 KO paired with latency reversal agents (LRAs) increases occupancy of elongation-competent RNAPII throughout the provirus. CUT&Tag using antibodies to INTS12 (**A**), total RNAPII (RPB3) (**B**), or RNAPII Ser2 phosphorylation (Ser2p) (**C**) were used to generate pileup graphs that show where each are binding chromatin. The location of where reads are mapping is denoted below the tracks, and the scale are the numbers on the left of each row. Each row represents three to five technical replicates averaged together. The 3' LTR is masked so that

*Figure 5 continued on next page*

*Figure 5 continued*

LTR-containing reads will only map to the 5' LTR. (**A**, left panel) shows a zoom-in of the HIV LTR and where INTS12 is binding. (**A**, right panel) quantifies total reads of INTS12 in the area denoted by the gray dashed lines in the left panel. (**B**, left panel) shows where RPB3 is binding across the HIV provirus and downstream. (**B**, right panel) quantifies total reads of RPB3 in the regions denoted by the gray dashed lines in the left panel, regions 1–3. (**C**, left panel) shows where RNAPII Ser2p is found across the HIV provirus and downstream. (**B**, right panel) quantifies total reads of RNAPII Ser2p in the regions denoted by the gray dashed lines in the left panel, regions 1–3. All chromosome locations and quantified regions can be found in the 'Materials and methods'. Two-tailed independent samples *t*-test *p-value<0.05, **<0.01, ***<0.001, ****<0.0001. Cells used in this figure have an INTS12 knockout score of 80%.

The online version of this article includes the following figure supplement(s) for figure 5:

**Figure supplement 1.** INTS12 localization with a second antibody.

**Figure supplement 2.** Ser5 phosphorylation localization.

the hypothesis that the combination of INTS12 with AZD5582 & I-BET151 specifically overcomes an elongation block that the INTS12 KO and AZD5582 & I-BET151 alone cannot overcome.

## INTS12 knockout improves latency reversal ex vivo in T cells from virally suppressed PLWH

We next tested the role of INTS12 in HIV-1 latency reversal in an ex vivo system of CD4 T cells isolated from PBMCs of virally suppressed PLWH. CD4 T cells were electroporated with Cas9 complexed with either a non-targeting control guide (NTC), or a cocktail of 3 INTS12 guides and then treated with DMSO (vehicle) or AZD5582 & I-BET151 treatment for 24 hr (*Figure 6A and B*). We used a low reactivating dose of AZD5582 (100 nM) & I-BET151 (500 nM) for primary cell experiments (*Falcinelli et al., 2022*). We tested six different donors with or without AZD5582 & I-BET151 treatment and measured cell-associated viral RNA expression by qPCR using primers in *gag* (*Figure 6A and B*). In *Figure 6A*, when we treat donors with the vehicle control (DMSO), we find that INTS12 KO trends upward for viral RNA expression in all six donors and this increase is statistically significant in four donors (*Figure 6A*). These same samples were treated with AZD5582 & I-BET151 (*Figure 6B*), and we found that although LRAs alone could activate viral RNA expression (compare samples in *Figure 6A* with *Figure 6B*), the combination of AZD5582 & I-BET151 paired with the INTS12 KO shows an even further increase in HIV-1 reactivation that was statistically significant in four out of six donors (*Figure 6B*). For donor PH543 (upside-down triangle), we found that there is an increase from the INTS12 KO itself (*Figure 6A*, right panel), and robust activation from AZD5582 & I-BET151 treatment itself (*Figure 6B*, right panel); however, there was no further increase from INTS12 KO with AZD5582 & I-BET151 treatment. Degree of reactivation was correlated with reservoir size as donors PH504 (star symbol) and PH543 (upside-down triangle) have the largest HIV reservoirs (*Supplementary file 4*). This data shows that our screen was able to identify INTS12 as a barrier to HIV-1 reactivation in PLWH as we find that there is reactivation from INTS12 knockout alone and improvement of AZD5582 & I-BET151 treatment with the addition of INTS12 knockout in a majority of virally suppressed PLWH tested ex vivo.

Because elongation of full-length HIV RNA has been a barrier to reactivation in PLWH previously, we not only wanted to look within the cells at *gag* RNA expression, but we sought to measure viral RNA in the supernatant by qPCR as a proxy for virus production. For this repeat assay, we chose three donors that had available materials (labeled in green in *Figure 6*). We used the same LRA concentrations we used in *Figure 6A and B*, but extended the time of treatment to 72 hr and measured viral RNA in the supernatant (*Figure 6C*). Importantly, we observed for all three donors that INTS12 KO was able to improve reactivation either on its own or in the presence of AZD5582 & I-BET151. Excitingly, donor 543 (*Figure 6C*, right panel) showed a robust increase from the pairing of INTS12 KO with AZD5582 & I-BET151 treatment when measuring gag RNA in the supernatant. The differences between *Figure 6C and B* for donor 543 could be due to the timing of the assay, or alternatively, pairing INTS12 KO with AZD5582 & I-BET151 results in more efficient full-length viral RNA such that we do not see an increase within the cell with the addition of INTS12 KO (*Figure 6B*) because there is an increased amount of RNA into the supernatant (*Figure 6C*, difference between LRA with or without INTS12 KO). Nevertheless, we can conclude that the majority of these donors are responsive to INTS12 KO with or without AZD5582 & I-BET151 treatment and that this reactivation leads to full-length HIV that can be detected in the supernatant.

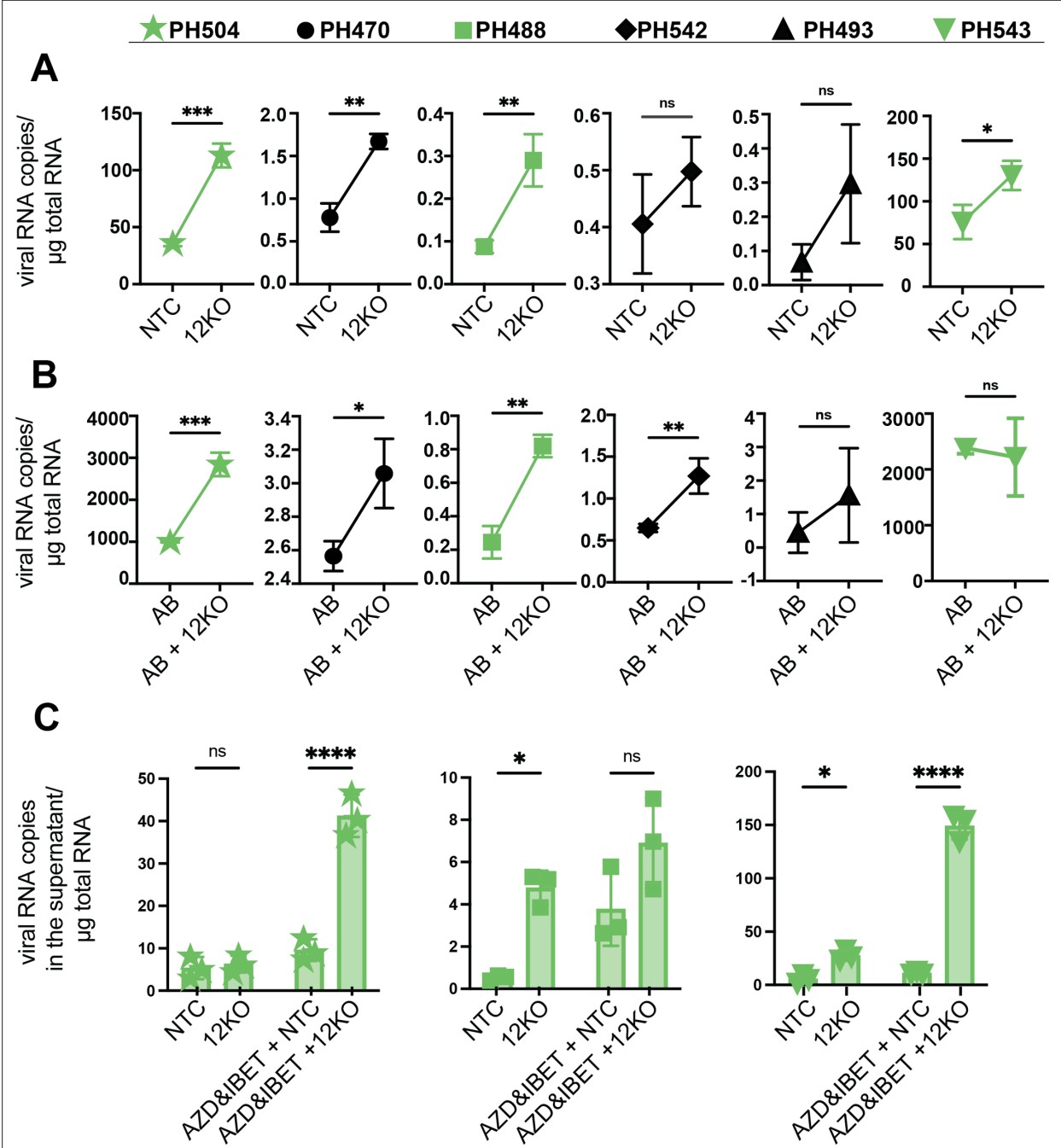

**Figure 6.** INTS12 knockout in an ex vivo primary cell system shows reactivation from knockout alone and in combination with AZD5582 & I-BET151. The number of gag copies/ug of RNA was measured from the cell (**A, B**) or the supernatant (**C**) for different donors, marked by different symbols, above. (**A**) INTS12KO (12KO) was compared to a non-targeting control (NTC). (**B**) AZD5582 & I-BET151-treated INTS12 KO (AB + 12 KO) was compared to AZD5582 & I-BET151 treatment (AB). n=3 replicates for each condition. Unpaired *t*-test, *p-value<0.05, **<0.01, ***<0.001, ****<0.0001. The green coloration denotes samples that are in common between (**A–C**). (**C**) Viral RNA in the supernatant was measured for three different donors. n=3 replicates for each condition. Two-way ANOVA *p-value<0.05, **<0.01, ***<0.001, ****<0.0001.

## The majority of the Integrator complex contributes to a block to reactivation and INTS12 can be targeted to increase HIV reactivation with a broad spectrum of stimuli

Previous studies have shown that Integrator members are important for HIV latency (*Stadelmayer et al., 2014*), and some have done so in the context of combination LRA treatment (*Li et al., 2020*).

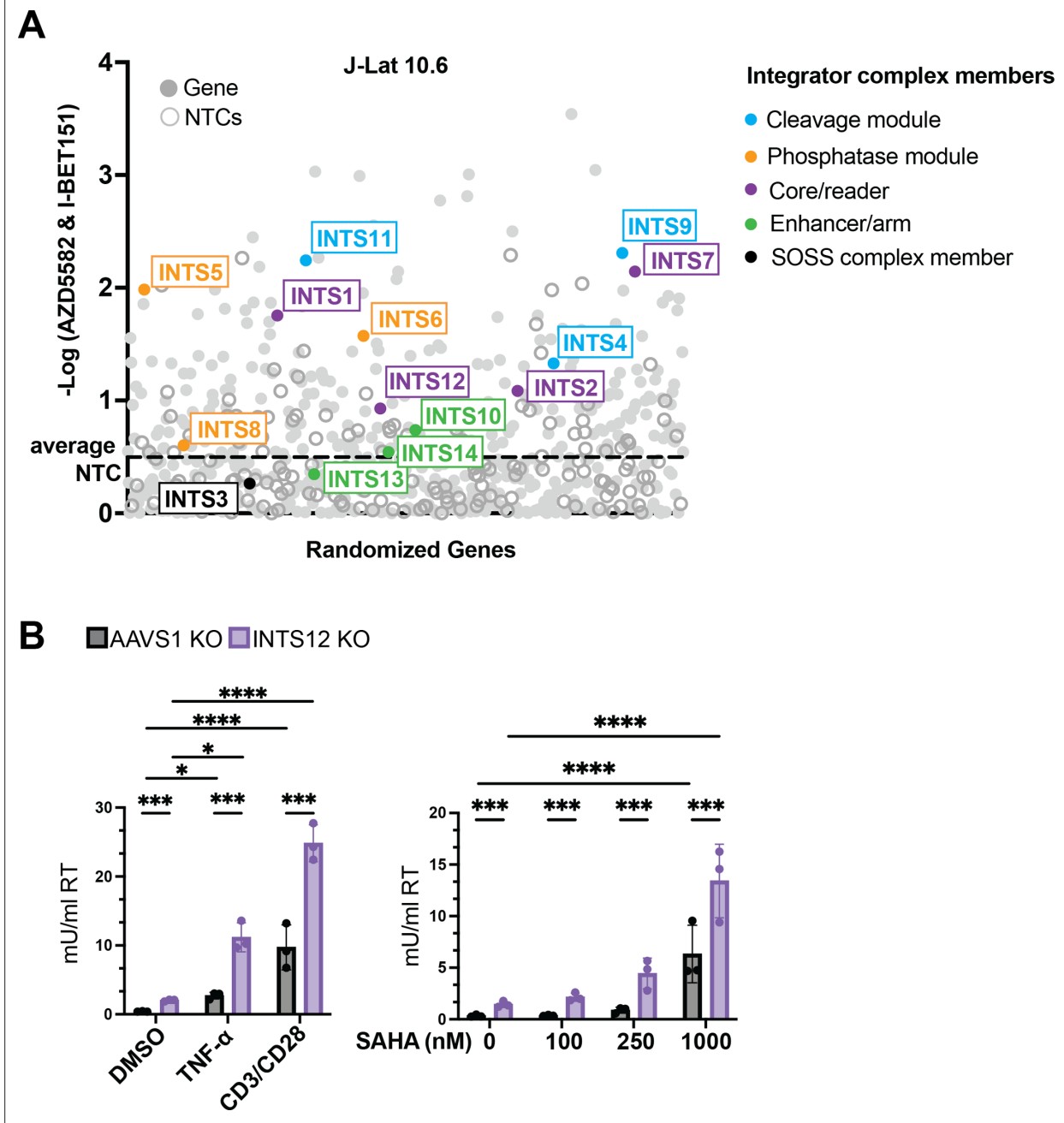

**Figure 7.** Most of the Integrator complex contributes to a block to HIV reactivation and INTS12 can be targeted to increase HIV reactivation with a broad spectrum of stimuli. (**A**) Screen #2 was performed in J-Lat 10.6 cells of a library of NF-kB-related factors where all Integrator complex members were added. The y-axis is the -log (MAGeCK gene score) that denotes how enriched a gene is. INTS12 has been shown to bind INTS1 (*Fianu et al., 2021*), so INTS12 was included in the core group coloration. The average NTCs are marked by a dotted line. (**B**) J-Lat 10.6 cells were knocked out for INTS12 or control locus, AAVS1. Cells were then treated with TNF-α or stimulated with anti-CD3/anti-CD28 antibodies for 20 hr (left panel), or SAHA for 24 hr (right panel). HIV reverse transcriptase activity was measured from the supernatant (reported in mU/mL). For statistical analysis, all conditions are compared to the AAVS1 control. n=3 replicates. Two-way ANOVA, (**B**) uncorrected Fisher's LSD, *p-value<0.05, **<0.01, ***<0.001, ****<0.0001. All cells used have an INTS12 knockout score of 75%.

In our initial screen of epigenetic factors (*Figure 1*), only INTS12 was included in the CRISPR guide library of the 15-member Integrator complex. In order to more comprehensively interrogate the role of Integrator complex members in HIV latency, we remade a library of guides in the HIV-CRISPR vector where we included all Integrator members (with the exception of INTS15 that was not known at the time of library construction *Offley et al., 2023*; *Replogle et al., 2022*). The remainder of this library

targets NF-kB-related factors (**Supplementary file 5**). We took this new library (herein, Screen #2) and treated J-Lat 10.6 cells with 1 nM AZD5582 & 2.5 μM I-BET151 for 24 hr (**Figure 7**) and found INTS12 is again predicted to improve the AZD5582 & I-BET151 treatment as it falls above the average NTCs. Importantly, however, all members of the cleavage module: INTS11, INTS9, INTS4; all members of the phosphatase module: INTS5, INTS6, INTS8; and all members of the core: INTS1, INTS2, INTS7 score positively in our screen. Thus, we predict that a large majority of the Integrator complex, consisting of the cleavage module, phosphatase module, core, and INTS12, but not the SOSS complex or the enhancer/arm module, likely play a role in HIV latency reversal in the presence of LRAs.

Given that the Integrator complex acts on the late elongation stage in viral transcription, it is possible that it serves as a more general block to HIV reactivation, and therefore, we would expect that INTS12 knockout could improve an even broader subset of LRAs than AZD5582 & I-BET151. We took our AAVS1 KO cells or INTS12 KO cells and reactivated with TNF-α, a canonical NF-kB activator, or anti-CD3/anti-CD28 co-stimulation that activates T cells through the T cell receptor and observed that INTS12 KO is able to increase reactivation with both of these stimuli (**Figure 7B**, left panel). Additionally, we tested these same cells with SAHA, an HDAC inhibitor that falls within the chromatin modification LRA class, for 24 hr, and we observe greater reactivation in our 250 nM and 1000 nM SAHA-treated INTS12 KO cells compared to our AAVS1 KO cells (**Figure 7B**, right panel). Therefore, INTS12 knockout was able to increase reactivation with all LRAs tested, which supports the idea that INTS12 serves as a general block to reactivation from latency for HIV.

## Discussion

We sought to improve HIV reactivation with a synergistic LRA combination AZD5582 & I-BET151 using a CRISPR screening approach. The ideal gene target would not only increase potency of this drug combination but maintain specificity to the virus and limit off-target effects. We used an epigenetic regulatory factor-focused guide library in combination with AZD5582 & I-BET151 and identified INTS12 as the top hit. We validated that knockout of INTS12 results in increased latency reversal on its own but more in the presence of AZD5582 & I-BET151 treatment, as the screen predicted. We observed that INTS12 KO resulted in specific enrichment of HIV and minimal effects on host genes, and that INTS12 KO is more specific to HIV than AZD5582 & I-BET151 treatment alone. INTS12 KO paired with AZD5582 & I-BET151 treatment further increased potency of HIV reactivation, but some specificity was lost compared to AZD5582 & I-BET151 alone and INTS12 KO alone. However, INTS12 KO paired with AZD5582 & I-BET151 remains more specific to HIV than PMA and ionomycin combination treatment. With CUT&Tag, we observed that INTS12 is at the HIV promoter which supports that it is acting directly on HIV-1 transcription, and we see an increase in elongation-competent RNAPII in the gene body of HIV only with the combination of INTS12 KO and AZD5582 & I-BET151, supporting that an elongation block is specifically overcome with this combination. Additionally, we validated INTS12 in an ex vivo primary cell model of latency where we see reactivation from INTS12 KO in a majority of donors tested with or without AZD5582 & I-BET151 treatment, and that reactivation upon INTS12 KO leads to detection of viral RNA in the supernatant, indicating that full-length HIV is likely generated. Furthermore, we used the CRISPR-screen approach to predict which members of the Integrator complex could be targeted to improve the AZD5582 & I-BET151 combination and identified the cleavage module, phosphatase module, and core Integrator members, in addition to INTS12 as playing a role in HIV-1 latency. Lastly, we demonstrated that INTS12 knockout can improve reactivation of a diverse set of stimuli supporting that INTS12 serves as a general block to reactivation of HIV from latency. This study highlights the promise of using the CRISPR-screening approach as a tool to identify blocks to HIV reactivation and improve LRA combinations, such as AZD5582 & I-BET151.

### The role of Integrator complex in modulating HIV transcription

The Integrator complex is known to regulate and attenuate transcription at promoters that are bound by pausing factors NELF and DSIF (**Skaar et al., 2015**). HIV is known to be regulated by these pausing factors (**Ping and Rana, 2001**), and it has been shown that members of the Integrator are recruited to the HIV TAR loop and regulate processivity of HIV transcription (**Stadelmayer et al., 2014**). The Integrator contains 15 members that fall within different submodules that perform various functions (**Welsh and Gardini, 2023**), and of these modules some members of the cleavage, phosphatase, and

core have been shown to reverse HIV latency upon knockout (*Li et al., 2020*; *Stadelmayer et al., 2014*). Conversely, knockout of INTS3 of the SOSS complex was not found to reverse latency in a previous study (*Stadelmayer et al., 2014*). Furthermore, members of the core and phosphatase have been shown to improve an LRA combination of a BET inhibitor with a PKC agonist (*Li et al., 2020*). We have recapitulated these previous findings (*Figure 7*), but we have also predicted that all members of these modules can be targeted to improve the AZD5582 & I-BET151 treatment, including the cleavage module for which this has not been explored in the context of LRAs.

One study explored the different effects on gene transcription upon targeting the cleavage module versus targeting the phosphatase module of the Integrator complex (*Hu et al., 2023*). Selectively targeting the cleavage module of Integrator has been shown to result in a specific induction of lowly expressed genes since these genes are regulated by pausing factors and Integrator (*Hu et al., 2023*). Given that HIV is regulated by these same factors and poorly expressed during HIV latency, it may not be surprising that targeting this module has shown an increase in HIV transcription. Since the Integrator complex has been shown to modulate genes differently in the presence or absence of stress stimuli (*Tatomer et al., 2019*; *Yue et al., 2017*), this may explain why targeting the Integrator complex in the presence of LRAs can improve reactivation. Moreover, Integrator is known to antagonize host P-TEFb function with its phosphatase module and enforce pausing at bound promoters. Therefore, targeting the phosphatase module can lead to more elongation at affected promoters. Given that HIV is sensitive to pausing and elongation, especially in cells from PLWH, and P-TEFb is considered to be one of the major rate-limiting steps to HIV reactivation, it is reasonable that removing a P-TEFb antagonist would help aid in HIV reactivation. Our results are also consistent with the hypothesis that transcription elongation with INTS12 KO cells treated with AZD5582 & I-BET151 may be more efficient than the LRAs alone as the levels of Ser2 phosphorylation of the C-terminal tail of RNAPII are also significantly lower near the LTR, but increased distal to the LTR (*Figure 5C*, compare INTS12 KO treated with AZD5582 & I-BET151 to DMSO in regions 1 and 2), which may suggest RNAPII is efficiently leaving the promoter and traveling within the body of the provirus to affect full-length transcription. This full-length transcription is also consistent with increased virus release that we observe both in cell lines and in primary cells (*Figures 2 and 6C*).

## INTS12 as a target for small-molecule drug development

While our data suggests that many different members of the Integrator complex could be targeted to activate HIV-1 (*Figure 7A*), we believe that INTS12 may be the better option to target over the rest of the complex. We find that INTS12 is located at the HIV-1 promoter (*Figure 5A*) and knockout fairly specifically reactivates HIV compared to host genes (*Figure 4A*, middle panel), which supports that INTS12 acts directly on HIV-1 transcription rather than indirectly through the modulation of other genes. There are several hypotheses for how targeting INTS12 is affecting HIV transcription. It could be that targeting INTS12 may keep catalytic activities of the Integrator complex partially intact or alter where those activities are exerted as this member is thought to toggle chromatin localization of the Integrator complex. Alternatively, it is possible INTS12 may be performing a separate function entirely from the remainder of the Integrator complex. In support of INTS12 being the best candidate to target, other studies have suggested that INTS12 knockdown does not grossly affect cell viability (*Kheirallah et al., 2017*), as opposed to other members of the Integrator complex (*Rienzo and Casamassimi, 2016*), although it has been shown that even catalytic members have been able to be targeted for short periods of time without signs of catalytic activity disruption in cell lines (*Elrod et al., 2019*). Therefore, with the caveats that the function of INTS12 in diverse T cell functions or other in vivo functions is not yet known and needs to be investigated in humans, we propose that targeting INTS12 with a small-molecule drug may be even more successful than targeting other Integrator members with short-term disruption in the shock-and-kill approach.

## Ex vivo primary cell system

To our knowledge, our study is the first to report an analysis of an Integrator complex member in an ex vivo experiment in cells from virally suppressed PLWH. We observed that INTS12 knockout reactivates HIV transcription in a majority of donors on its own and especially in the presence of AZD5582 & I-BET151 (*Figure 6*). This was in people with differing reservoir sizes, cells that have different HIV integration sites, and people on ART treatment for varying lengths of time. We were also able to show

that HIV RNA could be detected in the supernatant, which is an aspect critical to consider with regard to the kill side of the shock-and-kill approach and has proven to be one of the more difficult elements of the approach so far with many studies showing reactivation but variable virus production. This result highlights the power in the CRISPR-screening approach and shows that even when cell line models are used, we can use LRA combinations and identify blocks to reactivation that can be used to improve reactivation in more clinically relevant cells from PLWH.

## Synergy on multiple mechanisms

AZD5582 & I-BET151 are thought to synergize by acting to undo different blocks to latency reversal. AZD5582 working on the transcription initiation arm where noncanonical NF-kB is upregulated and I-BET151 on the transcription elongation arm where it is thought to impact available cellular P-TEFb that can work in conjunction with HIV Tat for HIV transcription (*Jang et al., 2005*; *Schröder et al., 2012*; *Turner et al., 2024*; *Yang et al., 2005*). When Integrator is bound to the HIV promoter, it can attenuate HIV transcription through its cleavage and phosphatase modules, as evidenced by knockout of members of these modules. As such, Integrator works on the elongation stage of transcription, and in our work, we have evidence that knocking out INTS12 overcomes a block to elongation. We not only see virus in the supernatant upon INTS12 KO in cells from PLWH (*Figure 6C*), but we also see that specifically with the combination of INTS12 KO and AZD5582 & I-BET151 there is an increase in elongation-competent RNAPII downstream of the HIV promoter and throughout the HIV provirus (*Figure 5B and C*). Therefore, our hypothesis is that targeting Integrator may allow for HIV transcription initiation events to not be aborted prematurely and allow for subsequent paused promoters to proceed to elongation from an unopposed P-TEFb. For this reason, removal of Integrator can lead to latency reversal on its own, but this also may explain why Integrator increases latency reversal with each LRA individually as well as improves the combination of INTS12 knockout with AZD5582 & I-BET151.

The Integrator complex itself is a fascinating target that needs to be considered for any HIV cure approach involving transcription. Not only does it govern transcription regulation at the HIV promoter (*Figure 3*), but it appears to do so in ways that may be contrary to the ways other genes are affected given its relative specificity for HIV transcripts (*Figure 4A*, middle panel), and has the ability to influence reactivation with diverse LRAs. While it has been reported that Integrator can function differently in the presence of stress stimuli (*Tatomer et al., 2019*; *Yue et al., 2017*), we did not observe differential expression of any Integrator members upon AZD5582 & I-BET151 treatment in J-Lat cells (*Supplementary file 3*, tabs 1 and 5) though this does not mean that recruitment of the complex does not vary under stimulatory conditions. In *Figure 5*, we did not observe significantly different recruitment of INTS12 to the HIV LTR upon AZD5582 & I-BET151 treatment but did not look at recruitment of other Integrator members and how their recruitment may be impacted. It is possible the general stoichiometry of the various Integrator members is important for the function of the complex at various promoters, though this has not been explored. Understanding how the Integrator complex and its modules are affected by different LRA stimuli may be crucial to optimizing the shock-and-kill strategy or even improving the reverse strategies of block-and-lock (*Lyons et al., 2023*) seeking to silence HIV more effectively at the HIV promoter. While, to our knowledge, small-molecule drugs that target INTS12, or any member of the Integrator complex are not yet available, the development of such agents would be predicted to improve the efficacy of existing LRAs to achieve more effective viral reactivation as a step toward reservoir reduction.

## Materials and methods

### Cell culture and maintenance

J-Lat 10.6 and J-Lat 5A8 cells were cultured with RPMI 1640 media (Thermo Fisher, 11875093) supplemented with 10% fetal bovine serum (FBS), Penicillin/Streptomycin, and 10 mM HEPES (Thermo Fisher, 15630080). HEK293T and TZM-bl cells were cultured in DMEM (Thermo Fisher, 11965092) supplemented with Penicillin/Streptomycin and 10% FBS. All cells were maintained at 37°C with 5% $CO_2$.

### CRISPR screen: Human Epigenetic (HuEpi) Library

The HuEpi screen was performed as described in *Hsieh et al., 2023*. Previously, J-Lat 10.6 and J-Lat 5A8 cells were prepared for the HIV-CRISPR latency screen by knocking out zinc antiviral protein

(ZAP), and the Human Epigenome Guide Library (HuEpi) was cloned into HIV-CRISPR vectors. Specific details for this article below.

## HuEpi screen: Lentivirus production

Lentivirus was made by adding 667 ng of previously prepped HuEpi HIV-CRISPR vectors (*Hsieh et al., 2023*), 500 ng psPAX2 (GagPol), and 333 ng MD2.g (VSVG) per well in 200 uL of serum-free DMEM and completed with 4.5 µL of TransIT-LT1 reagent (Mirus Bio LLC, MIR 2304). 20 × 6 well cell culture plates of 293Ts were used at a confluency of 40–60%. Media was replaced 24 hr after transfection (and reduced to 1.5 mL/well) and the VSV-G pseudotyped lentivirus was harvested 48 hr after that and filtered through a 0.22 µM filter (Thermo Scientific; 720-1320). Virus supernatant was combined (by cell type) and concentrated by ultracentrifugation. For ultracentrifugation, about 30 mL of supernatant was aliquoted into each polypropylene tube (Beckman Coulter; 326823) and sterile-filtered 20% sucrose (20% sucrose, 1 mM EDTA, 20 mM HEPES, 100 mM NaCl, distilled water) was added slowly underneath to form a sucrose cushion. Tubes were then placed in a pre-chilled swinging buckets SW 28 rotor at 23,000 rpm for 1 hr at 4°C in a Beckman Coulter Optima L-90K Ultracentrifuge. Supernatants were decanted, allowed to dry briefly, and pellets were resuspended overnight in 600 µL RPMI at 4°C (can do several hours). Concentrated lentivirus was then stored at –80°C in single-use small aliquots (~150 µL) and two 50 uL aliquots were stored at –80°C for titration.

## HuEpi screen: HuEpi HIV-CRISPR cell generation

Virus was titered on TZM-bl cells (NIH AIDS Reagent Program; ARP-8129) in a colony-forming assay to calculate for a <1 multiplicity of infection (MOI) (0.54 was used in the HuEpi screens). The number of cells to transduce was calculated to be >500x (5029 HuEpi guides × 500 = 2.51e6 + error = 3e6 cells), so 3e6 J-Lat cells (ZAP KOs) were transduced per replicate. Cells were transduced with DEAE-Dextran (20 µg/mL final concentration, Sigma-Alrdich; D9885) and spinoculated for 30 min at 1100 × *g* and incubated overnight. After 24 hr, the supernatant was replaced with fresh media containing 0.4 µg/mL of puromycin (Sigma) and cells were selected for 10–14 days (cells were grown for 21 days in this experiment to acquire enough for the desired screen coverage and all experimental conditions).

## LRA resuspension

AZD5582 (MedChemExpress, HY-12600), I-BET151 (SelleckChem, S2780) TNF-α (Peprotech, 300-01A), SAHA (Selleckchem, S1047), and Tazemetostat (Selleckchem, S7128) were resuspended in DMSO to make 10 mM stocks stored at –80°C. Stocks were then diluted down to single-use working concentrations for experiments with RPMI.

## T-cell activation

Anti-human CD3 antibody (Tonbo Biosciences, 40-0038-U500) and anti-human CD28 antibody (Tonbo Biosciences, 40-0289-U500).

## HuEpi screen: LRA treatments

For the LRA treatments in the screen, we use a coverage of 5000× in case reactivation events are rare and we also account for viability. We calculated 25e6 cells were needed per replicate + error = 26e6 cells. We accounted for AZD5582's viability and calculated 35e6 cells/replicate for each condition. AZD5582 & I-BET151 were resuspended in DMSO to 1 mM stocks stored at –80°C. All other dilutions were performed in RPMI and used immediately. For the screen, 35e6 cells/replicate were grown at 5e5 cells/mL in a flask and treated with 10 nM AZD5582, 100 nM I-BET151, 10 nM AZD5582 & 100 nM I-BET151, or DMSO. Cells and supernatants were collected after 48 hr of stimulation. Cells were spun down at 1500 × *g* for 3 min, supernatants were transferred to a new 50 mL conical, filtered through a 0.22 µm filter (MilliporeSigma, SE1M179M6), and stored at 4°C (can store overnight) for virus concentration. Cells were washed once with DPBS (Gibco; 14190144) and resuspended in 1 mL DPBS to store aliquots of 5e6 cells (approximately calculated, not counted) at –80°C. Finish concentrating virus by following steps detailed in lentivirus production (above) but resuspend the pellet in 150 µL at the end. Freeze viral supernatant at –80°C.

### HuEpi screen: Sample processing and prepping for sequencing

Genomic DNA from cell pellets were extracted with the QIAamp DNA Blood Midi Kit (QIAGEN; 51183) and eluted in distilled water. vRNA from viral supernatant was extracted with the QIAamp Viral RNA Mini Kit (QIAGEN, 52904). PCR 1: sgRNA sequences are in the gDNA and vRNA samples, and they are amplified by PCR (Agilent; 600677) and RT-PCR (Invitrogen; 18064014), respectively, using HIV-CRISPR-specific primers. PCR 2: adds barcodes and prepares the libraries for Illumina sequencing (examples of barcodes in *Hsieh et al., 2023*). Each amplicon was cleaned up using double-sided bead clean-up (Beckman Coulter; A63880), quantified with a Qubit dsDNA HS Assay Kit (Invitrogen; Q32854), and pooled to 10 nM for each library. Library pools are sequenced on a single lane of an Illumina HiSeq 2500 in Rapid Run mode (Fred Hutch Genomics and Bioinformatics shared resource).

### HuEpi screen: Screen analysis

Analysis was performed as described in *Hsieh et al., 2023*. Briefly, sequencing reads were demultiplexed and assigned to samples, trimmed, and aligned to the HuEpi library with Bowtie. Guide enrichment or depletion was determined using the MAGeCK statistical package.

### Screen #2 containing all Integrator complex members

This screen was performed similarly to the HuEpi screen described above, with the following changes. The library contains 517 genes (6 guides/gene) + 163 NTCs = 3265 guides total. 1 nM AZD5582 & 2.5 µM I-BET151 was used for LRA treatment at 24 hr. Coverage during transduction was 500×, and coverage during LRA treatments was 3600×. 50 bp sequencing was performed on the MiSeq V3 platform by the Fred Hutchinson Cancer Research Center Genomics Shared Resources.

### CRISPR KOs/electroporations

J-Lat cells (in *Figures 2A, 3, and 4*) were knocked out for INTS12 or AAVS1 with single guides cloned in the HIV-CRISPR. Transduced cells were selected with 0.4 µg/mL puromycin selection for 10–14 days to generate CRISPR/Cas9-edited knockout pools. J-Lat cells (in *Figure 2B and C*) were knocked out for INTS12 using a cocktail of three guides from IDT (Predesigned Alt-R CRISPR-Cas9 guide RNA) or AAVS1 using a cocktail of three guides from Synthego (ordered individually as gene knockout kit v2 and then pooled together) and Cas9 from IDT (Alt-R S.p. Cas9 Nuclease V3). Cells were electroporated using the SE Cell Line 4D-NucleofectorTM X Kit L from Lonza using the CL-120 pulse code and edited for at least 3 days before use. All guides used are located in *Supplementary file 2*.

### Vector for complementation

For the complementation experiment (*Figure 2B and C*), after cells were electroporated and knocked out with the cocktails of three guides detailed above, half of the cells were transduced with a vector containing INTS12 generated on VectorBuilder (pLV[Exp]-Puro-EF1a>hINTS12[NM_020395.4]) and selected with 0.4 µg/mL puromycin for 10 days before LRA treatments. Cells were double-checked for knock by ICE analysis and knockout scores only dropped by ~5% during this selection period.

### ICE analysis

PCR products of a region containing the edited site in INTS12 knockout cells and not edited site in AAVS1 knockout cells were compared using the Synthego ICE analysis software to generate a knockout score. All primers can be found in *Supplementary file 2*.

### LRA treatments

AAVS1 KO and INTS12 KO cells were seeded at a density of 5e5 cells/mL in 96-well plates with 200 uL/well or in flasks. LRA treatments were performed in triplicate. Cells were seeded and then the calculated amount of LRA was added on top with mixing and cells were incubated. All J-Lat experiments used 10 nM AZD5582 & 100 nM I-BET151 for 48 hr except Screen 2 (*Figure 7*) used 1 nM AZD5582 & 2.5 uM I-BET151 for 24 hr. Primary cell experiments used 100 nM AZD5582 and 500 nM I-BET151 for 24 hr for qPCR, and then treatment was extended to 72 hr for viral RNA detection in the supernatant. Cells were treated with 10 ng/mL TNF-α for 20 hr and SAHA for a concentration between 1000 nM to 100 nM for 24 hr.

## T-cell activation protocol

96-well plate wells were coated with 0.25 uL of anti-CD3 antibody in 50 uL PBS overnight at 4°C in a parafilmed plate overnight. Wells were aspirated and 200 uL of cells at a density of 5e5 cells/mL were added to the coated well. Anti-CD28 antibody was added to a concentration of 2 mg/mL to the final well. This stimulation went on for 20 hr before RT assay measurement.

## Virus measurement: RT assay

LRA-treated cells were spun at 1500 × $g$ for 3 min, and 50 μL of supernatant was transferred to 96-well plates. Plates were either processed immediately or were frozen at –80°C for future processing. For the RT assay, 5 μL of supe for each condition was mixed with lysis buffer and then diluted in $H_2O$ before use as template for qPCR. The detailed protocol can be found in *Vermeire et al., 2012*.

## RNA-seq run/analysis

Three biological triplicates were generated in J-Lat 10.6 cells for AAVS1 KO and INTS12 KO cells by knocking out three separate sets of cells, and then cells were treated with LRAs. We included AAVS1 treated with DMSO, AAVS1 treated with AZD5582 & I-BET151, INTS12 KO treated with DMSO, and INTS12 KO treated with AZD5582 & I-BET151. Treated samples were spun down at 1.45e6 cells/tube (I used two tubes/sample), the supernatant was aspirated, and cells were lysed with 700 μL of QIAzol (QIAGEN, 1023537). Samples were spun through a QIAshredder (QIAGEN, 79654) and then frozen at –80°C. Cells were processed with a miRNeasy kit (QIAGEN, 217004), DNase treated with RNase-Free DNase Set (QIAGEN, 79254), and then reprocessed through the RNeasy Mini Kit (QIAGEN, 74104) with the EtOH wash modified to 700 μL for small RNAs. Combine tubes for each sample. RNA quality was checked by TapeStation (all samples had a RIN of 10), and 10 μL of RNA at 50 ng/μL was sent for RNA-seq to the Fred Hutchinson Cancer Center Genomics Shared Resources where rRNA was removed to select for the rest of the total RNA with the Illumina Stranded Total RNA Prep, Ligation with Ribo-Zero Plus kit (20040525) followed by indexing with IDT for Illumina RNA UD Indexes Set A Ligation (20040553). Samples were run using paired-end 2 × 50bp sequencing on the NovaSeq S1 by the Fred Hutchinson Cancer Center Genomics Shared Resources.

## RNA-seq: Data analysis

Fastq files were aligned to the custom genome assembly prepared by *Hsieh et al., 2023* in which chr9:136468439–136468594 of hg38 is replaced with the integrated HIV-1 sequence from MN989412.1 (*Chung et al., 2020*), and the second 634 bp copy of the LTR is masked by 'N's. GENCODE annotation v38 and HIV gene annotation were combined and genome coordinates were modified accordingly. STAR v2.7.7a (*Dobin et al., 2013*) with two-pass mapping was used to align paired-end reads to the custom reference. FastQC 0.11.9 (available here), RNA-SeQC 2.3.4 (*DeLuca et al., 2012*), and RSeQC 4.0.0 (*Wang et al., 2012*) were used to check various QC metrics, including insert fragment size, read quality, read duplication rates, rRNA rates, gene body coverage, and read distribution in different genomic regions. featureCounts (*Liao et al., 2014*) in Subread 2.0.0 was used to quantify gene-level read counts in second-stranded fashion. Bioconductor package edgeR 3.36.0 (*Robinson et al., 2010*) was used to detect differential gene expression between sample groups. Genes with low expression were excluded using edgeR function filterByExpr with min.count = 10 and min.total.count = 15. The filtered expression matrix was normalized by TMM method (*Robinson and Oshlack, 2010*) and subject to significance testing using quasi-likelihood pipeline implemented in edgeR. A gene was deemed differentially expressed if absolute log2 fold change was above 1 (i.e., fold change >2 in either direction), and Benjamini–Hochberg adjusted p-values was <0.05.

## CUT&Tag: Sample preparation

Nuclei were prepared from J-Lat 10.6 cells. Samples included AAVS1 treated with DMSO, AAVS1 treated with AZD5582 & I-BET151, INTS12 KO treated with DMSO, and INTS12 KO treated with AZD5582 & I-BET151. Treated cells were pelleted in 1.5 mL microfuge tubes (10e6 cells/sample) spun at 300 × $g$ for 5 min, and then resuspended in 1 mL of ice-cold NE1 buffer (20 mM HEPES-KOH pH 7.9, 10 mM KCl, 0.5 mM spermidine, 0.1% TritonX-100, 20% glycerol, with Roche complete EDTA-free protease inhibitor tablet), and then incubated on ice for 10 min. Nuclei were centrifuged at 4°C at 1300 × $g$ for 4 min and then resuspended in 1 mL of Wash Buffer (20 mM HEPES pH 7.5, 150 mM

NaCl, 0.5 mM spermidine, supplemented with Roche complete EDTA-free protease inhibitor tablet). Nuclei were counted (10 µL was used) and diluted to a concentration of 1e6 nuclei/900 µL of wash buffer and aliquoted into cryovials with 100 µL of DMSO, which were quickly placed inside a Mr. Frosty Isopropanol chamber for slow freezing at –80°C. Nuclei were then stored at –80°C until use. For automated CUT&Tag processing, nuclei were thawed at room temperature, washed in wash buffer, and bound to concanvalin-A (ConA) paramagnetic beads (Bangs Laboratories; BP531) for magnetic separation as described on the protocols.io website (https://doi.org/10.17504/protocols.io.bgztjx6n). Samples were then suspended in antibody binding buffer and split for overnight incubation with antibodies specific to INTS12 (Thermo Fisher, 16455-1-AP) (*Figure 5A*) or INTS12 (Abbexa, abx234360) (*Figure 5—figure supplement 1*) or RPB3 (Thermo Fisher, A303-771A), RNA-Pol2-S5p (Cell Signaling; 13523), RNA-Pol2-S2p (Cell Signaling; 13499), and IgG control (Abcam; 172730). Sample processing was performed in a 96-well plate using 25K Con-A bound nuclei per reaction on a Beckman Coulter Biomek liquid handling robot according to the AutoCUT&Tag protocol available from the protocols.io website (https://doi.org/10.17504/protocols.io.bgztjx6n) and described previously (*Janssens et al., 2022*) (Fred Hutch Genomics Shared Resources). Samples were run on the NextSeq P2 using paired-end 2 × 50 bp sequencing by the Fred Hutchinson Cancer Research Center Genomics Shared Resources.

## CUT&Tag: Data analysis

The same custom genome assembly as RNA-seq was used for CUT&Tag. Bowtie2 version 2.4.2 was used for alignments with the following parameters: —very-sensitive-local—soft-clipped-unmapped-tlen—dovetail—no-mixed—no-discordant -q—phred33 -I 10 -X 1000. For each target protein and each condition, 3–5 technical replicates were profiled. We removed duplicate reads from each replicate, and the combined reads for all replicates were pooled and used to generate the coverage-normalized bigwig files shown as genome-browser tracks. The number of unique reads for each replicate was quantified over the intervals indicated by the dotted gray lines in *Figure 5*, and we compared the read counts over the indicated intervals in the AAVS1KO AZD&IBET, INTS12KO DMSO, and INTS12KO AZD&IBET conditions to the AAVS1KO DMSO control samples using a two-tailed independent samples *t*-test.

## CUT&Tag: Chromosome locations

*Figure 3B*: chr9 136,468,583–136478093; *Figure 5A*, right panel quantification: chr9 136,468,583–136,469,694; *Figure 5B and C*, right panel, region 1 quantification: chr9 136,468,583–136,469,694; *Figure 5B and C*, right panel, region 2 quantification: chr9 136,469,695–136,478,093; *Figure 5B and C*, right panel, region 3 quantification: 136,478,783–136, 487,890.

## Primary cells: Isolation

CD4 T cells were isolated from PBMC obtained from ART-suppressed PLWH and cultured in RPMI media supplemented with 10% FBS, Penicillin/Streptomycin, 10 mM HEPES, and 10 units per mL of IL-2. All cells from PLWH were obtained from the UNC clinical HIV cohort (study number: 08-1575).

## Primary cells: Knockout and LRA treatment

HIV-infected CD4 cells from PLWH in vivo were electroporated using the P2 Primary Cell 96-well Nucleofector Kit, (V4SP-2096, Lonza, Cologne, Germany), according to the manufacturer's protocol with slight modification (*Ashokkumar et al., 2024*): briefly, 2–3 × 10^6 cells were washed twice with phosphate-buffered saline by centrifuging at 90 × g for 5 min and resuspended in nucleofection buffer P2. The resuspended cells together with RNP complexes were immediately transferred into the cuvette of the P2 Primary Cell Nucleofector Kit (Lonza; V4SP-2096) and electroporated using the program code, EH-100 on the Lonza 4D-Nucleofector. Electroporated cells were resuspended with 200 µL of prewarmed supplemented RPMI media with IL-2 (10 U/mL) and IL-7 (4 ng/mL) and expanded in a 37°C incubator. Three days post nucleofection, cells were treated with combination of AZD5582 (100nM) and I-BET151 (500nM) or DMSO and the expression of cell-associated (24 hr) and cell-free (72 hr) gag was quantified by RT-qPCR. The cells were maintained at 1 × 10^6 cells/mL with fresh media supplemented with IL-2 and IL-7 every 2–3 days. Pre-designed crRNA with higher off-target score targeting human gene, INTS12 (CTTTTGAGATGGACGGTAAC, CACAATACCACACCAGGCGA,

and ACCAATGGATCTTTGACAGC), and/or scrambled guides (non-target control [NT]) were obtained from IDT as controls. Annealing of crRNA and tracrRNA, preparation of CRISPR/Cas9 ribonucleoprotein complexes (RNPs). Three sgRNA targeting different regions of the target were multiplexed for more efficient target knockout.

## Primary cell: RT-qPCR

Quantification of relative gene expression using quantitative-PCR One-step RT-qPCR HIV Gag RNA gene expression quantifications were performed as previously described *Falcinelli et al., 2022*. Briefly, RNA from independent donors were extracted using the RNeasy plus kit and QIAamp Viral RNA Mini Kit (QIAGEN) for cell-associated and cell-free virus as per the manufacturer's instructions, respectively. 1 ug of nanodrop quantified RNA from non-target (NT) or INTS12 gRNA nucleofected cells were reverse transcribed and amplified using Fastvirus (Thermo, Waltham, MA) and primer sets for HIV Gag RNA (GAG-F: ATCAAGCAGCCATGCAAATGTT, GAG-R: CTGAAGGGTACTAGTAGTTCCTGC TATGTC, GAGProbe: FAM/ZEN-ACCATCAATGAGGAAGCTGCAGAATGGGA-IBFQ). Reactions were performed in 96-well plates using the Quant Studio 3 Real-Time PCR system (Applied Biosystems, Foster City, CA) real-time thermocycler with a cycling parameters of 5 min reverse transcription step at 50°C, 95°C for 20 s for Taq activation, followed by 40 cycles of 95°C (3 s) and 60°C (30 s). All qPCR reactions were performed in triplicate. Normalized relative expression levels were calculated using the Prism software version 10.1.1 (GraphPad).

## Acknowledgements

We thank Steve Hahn and Toshi Tsukiyama for helpful discussions, Trine Ready for technical support, Jessie Kulsuptrakul for help with cell maintenance, Feinan Wu and Matthew Fitzgibbon for RNA-seq and CUT&Tag bioinformatics support, Alyssa Dawson and Alex Zevin for aid in RNA-seq set-up and sequencing, Merari Santana-Carbajal and Philip Corrin for aid in CUT&Tag set-up and sequencing, Harini Srinivasan for aid in constructing the library for Screen #2, Pritha Chanana for aid in analyzing Screen #2, Cindy Gay, Susan Pedersen, Caroline Baker, Taylor Whitaker and the staff of the UNC Blood Bank and UNC CTRC for clinical support, as well as all members of the Emerman lab and the CARE Collaboratory for helpful suggestions and discussions. We thank Warner Greene at Gladstone Institute of Virology and Immunology and University of California, San Francisco, San Francisco, CA, USA, for sharing the J-Lat 5A8 cells. We thank the HIV seropositive participants for their altruism. This work was supported by DP1 DA051110 (ME), CARE 1UM1-A1-164567 (ME, NMA, EPB), NIAID R01 AI143381 (EPB), NIDA R61 DA047023 (EPB), R01 AI143381 (EPB), 1R56AI170226 (NMA), and RRID:SCR_022606; additionally, of the Fred Hutch/University of Washington/Seattle Children's Cancer Consortium (P30 CA015704). DHJ was supported by a Hartwell Foundation Postdoctoral Fellowship. EH was supported by the University of Washington (UW) Viral Pathogenesis training grant NIH T32 AI 083203.

## Additional information

### Funding

| Funder | Grant reference number | Author |
| --- | --- | --- |
| National Institute on Drug Abuse | DP1 DA051110 | Michael Emerman |
| National Institute of Allergy and Infectious Diseases | 1UM1-A1-164567 | Nancie M Archin<br>Edward P Browne<br>Michael Emerman |
| National Institute of Allergy and Infectious Diseases | R01 AI143381 | Edward P Browne |
| National Institute on Drug Abuse | R61 DA047023 | Edward P Browne |

| Funder | Grant reference number | Author |
|---|---|---|
| National Institute of Allergy and Infectious Diseases | R56 AI170226 | Nancie M Archin |
| Hartwell Foundation | | Derek H Janssens |
| National Institute of Allergy and Infectious Diseases | T32 AI 083203 | Emily Hsieh |
| National Cancer Institute | P30 CA015704 | Michael Emerman |

The funders had no role in study design, data collection and interpretation, or the decision to submit the work for publication.

## Author contributions

Carley N Gray, Conceptualization, Data curation, Formal analysis, Validation, Investigation, Visualization, Methodology, Writing – original draft, Writing – review and editing; Manickam Ashokkumar, Investigation, Methodology, Writing – review and editing; Derek H Janssens, Software, Formal analysis, Investigation, Methodology, Writing – review and editing; Jennifer L Kirchherr, Brigitte Allard, Emily Hsieh, Terry L Hafer, Investigation, Methodology; Nancie M Archin, Formal analysis, Supervision, Funding acquisition, Project administration, Writing – review and editing; Edward P Browne, Funding acquisition, Writing – original draft, Project administration, Writing – review and editing; Michael Emerman, Conceptualization, Supervision, Funding acquisition, Writing – original draft, Project administration, Writing – review and editing

## Author ORCIDs

Carley N Gray https://orcid.org/0009-0008-5173-2769
Derek H Janssens https://orcid.org/0000-0003-1079-9525
Jennifer L Kirchherr https://orcid.org/0000-0003-3559-7356
Edward P Browne https://orcid.org/0000-0001-9070-7015
Michael Emerman https://orcid.org/0000-0002-4181-6335

## Ethics

PBMC for this study were obtained from durably suppressed on ART people living with HIV under a University of North Carolina (UNC) Institutional Review Board-approved protocol (study number 08-1575). All participants provided written consent.

Reviewer #1 (Public review): https://doi.org/10.7554/eLife.103064.3.sa1
Reviewer #2 (Public review): https://doi.org/10.7554/eLife.103064.3.sa2
Reviewer #3 (Public review): https://doi.org/10.7554/eLife.103064.3.sa3
Author response https://doi.org/10.7554/eLife.103064.3.sa4

# Additional files

## Supplementary files

Supplementary file 1. HuEpi Screen 1 results.

Supplementary file 2. Guides and ICE primers.

Supplementary file 3. RNA-seq files and Integrator complex expression compilation.

Supplementary file 4. PLWH reservoir information.

Supplementary file 5. NFkB-related Screen 2 containing all Integrator members.

MDAR checklist

## Data availability

All primary CUT&Tag sequencing data, RNA-seq data, and CRISPR screen data can be found at GSE277306. All primary values for figures found in Dryad https://doi.org/10.5061/dryad.qfttdz0t0.

The following datasets were generated:

| Author(s) | Year | Dataset title | Dataset URL | Database and Identifier |
|---|---|---|---|---|
| Gray C | 2025 | Integrator complex subunit 12 knockout overcomes a transcriptional block to HIV latency reversal | https://doi.org/10.5061/dryad.qfttdz0t0 | Dryad Digital Repository, 10.5061/dryad.qfttdz0t0 |
| Gray CN, Gray CN, Ashokkumar M, Janssens DH, Kirchherr J | 2024 | Integrator complex subunit 12 knockout overcomes a transcriptional block to HIV latency reversal | https://www.ncbi.nlm.nih.gov/geo/query/acc.cgi?acc=GSE277306 | NCBI Gene Expression Omnibus, GSE277306 |

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
