## [Editor Report · eLife Assessment]

Using multiple techniques previously validated by the authors, this study identified INTS12, a component of the Integrator complex involved in 3' processing of small nuclear RNAs U1 and U2, as a factor promoting HIV-1 latency. The work is **valuable**, based on a sound strategy for screening targets to activate HIV latency and the **solid** mechanistic insights it provides on INTS12 repression of transcriptional elongation. Future studies are needed to explore INTS12 as a drug target against HIV/AIDS.

---

## [Referee Report · Reviewer #1 (Public review)]

Gray and colleagues describe the identification of Integrator complex subunit 12 (INTS12) as a contributor to HIV latency in two different cell lines and in cells isolated from the blood of people living with HIV. The authors employed a high-throughput CRISPR screening strategy to knock down genes and assess their relevance in maintaining HIV latency. They had used a similar approach in two previous studies, finding genes required for latency reactivation or genes preventing it and whose knockdown could enhance the latency-reactivating effect of the NFκB activator AZD5582. This work builds on the latter approach by testing the ability of gene knockdowns to complement the latency-reactivating effects of AZD5582 in combination with the BET inhibitor I-BET151. This drug combination was selected because it has been previously shown to display synergistic effects on latency reactivation.

The finding that INTS12 may play a role in HIV latency is novel, and the effect of its knock down in inducing HIV transcription in primary cells, albeit in only a subset of donors, is intriguing.

In this revised version, the authors have included new data and clarifications which help strengthen their conclusions.

---

## [Referee Report · Reviewer #2 (Public review)]

Summary:

Identifying an important role for Integrator complex in repressing HIV transcription and suggesting that by targeting subunits of this complex specifically, INTS12, reversal of latency with and without latency reversal agents can be enhanced.

Strengths:

The strengths of the paper include the general strategy for screening targets that may activate HIV latency and the rigor of exploring the mechanism of INTS12 repression of HIV transcriptional elongation.

Weaknesses:

Minor point-there was an opportunity to examine a larger panel of latency reversal agents that reactivate by different mechanisms to determine whether INTS12 and transcriptional elongation are limiting for a broad spectrum of latency reversal agents.

Comments on revisions:

I feel the authors have adequately addressed the original questions and concerns.

---

## [Referee Report · Reviewer #3 (Public review)]

Summary:

Transcriptionally silent HIV-1 genomes integrated in the host`s genome represent the main obstacle for an HIV-1 cure. Therefore, agents aimed at promoting HIV transcription, the so-called latency reactivating agents (LRAs) might represent useful tools to render these hidden proviruses visible to the immune system. The authors successfully identified, through multiple techniques, INTS12, a component of the Integrator complex involved in 3' processing of small nuclear RNAs U1 and U2, as a factor promoting HIV-1 latency and hindering elongation of the HIV RNA transcripts. This factor hinders the activity of a previously identified combination of LRAs, one of which, AZD5582, has been validated in the macaque model for HIV persistence during therapy (https://pubmed.ncbi.nlm.nih.gov/37783968/). The other compound, I-BET151, is known to synergize with AZD5582, and is a inhibitor of BET, factors counteracting elongation of RNA transcripts.

Therefore, INTS12 maight represent a target for future LRAs-

Strengths:

Findings were confirmed through multiple screens and multiple techniques. The authors successfully mapped the identified HIV silencing factor at the HIV promoter, Silencing of INTS12 increases the activity of small-molecule HIV latency-reversing agents such as the histone deacetylase inhibitor vorinostat. Knockdown of INTS12 does not induce toxic effects in the cells, thus rendering it a candidate a drug discovery campaign aimed at finding new agents for an HIV/AIDS cure.

Weaknesses:

A caveat is that the impact of INTS12 in diverse T cell functions or other in vivo functions is not yet known, but the authors acknowledge this in the revised discussion.

---

## [Author Response]

The following is the authors’ response to the original reviews.

**Public Reviews:**

**Reviewer #1 (Public review):**
Gray and colleagues describe the identification of Integrator complex subunit 12 (INTS12) as a contributor to HIV latency in two different cell lines and in cells isolated from the blood of people living with HIV. The authors employed a high-throughput CRISPR screening strategy to knock down genes and assess their relevance in maintaining HIV latency. They had used a similar approach in two previous studies, finding genes required for latency reactivation or genes preventing it and whose knockdown could enhance the latency-reactivating effect of the NFκB activator AZD5582. This work builds on the latter approach by testing the ability of gene knockdowns to complement the latency-reactivating effects of AZD5582 in combination with the BET inhibitor I-BET151. This drug combination was selected because it has been previously shown to display synergistic effects on latency reactivation.The finding that INTS12 may play a role in HIV latency is novel, and the effect of its knockdown in inducing HIV transcription in primary cells, albeit in only a subset of donors, is intriguing. However, there are some data and clarifications that would be important to include to complement the information provided in the current version of the manuscript.

We have now added the requested data and clarifications. In particular, we show that knockout of INTS12 has no effect on cell proliferation (new data added in Figure 2—figure supplement 3), we clarify how the degree of knockout and the complementation were accomplished, we clarify the differences between the RNA-seq and the activation scores, and we have bolstered the claim that INTS12 affected transcription elongation by performing CUT&Tag on Ser2 phosphorylation of the C-terminal tail of RNAPII along the length of the provirus (new data added in Figure 5C) Please see detailed responses below.

**Reviewer #2 (Public review):**
Summary:Identifying an important role for the Integrator complex in repressing HIV transcription and suggesting that by targeting subunits of this complex specifically, INTS12, reversal of latency with and without latency reversal agents can be enhanced.Strengths:The strengths of the paper include the general strategy for screening targets that may activate HIV latency and the rigor of exploring the mechanism of INTS12 repression of HIV transcriptional elongation. I found the mechanism of INTS12 interesting and maybe even the most impactful part of the findings.Weaknesses:I have two minor comments:There was an opportunity to examine a larger panel of latency reversal agents that reactivate by different mechanisms to determine whether INTS12 and transcriptional elongation are limiting for a broad spectrum of latency reversal agents.I felt the authors could have extended their discussion of how exquisitely sensitive HIV transcription is to pausing and transcriptional elongation and the insights this provides about general HIV transcriptional regulation.

We have now added data on latency reversal agents of different mechanisms of action. We show that INTS12 affects HIV latency reversal from agents that affect the non-canonical NF-kB pathway (AZD5582), the canonical NF-kB pathway (TNF-alpha), activation via the T-cell receptor (CD3/CD28 antibodies), through bromodomain inhibition (I-BET151), and through a histone deacetylase inhibitor (SAHA). This additional data has been added to the manuscript in Figure 7, panels B and C as well as adding text to the discussion.

We appreciate the suggestion to extend the discussion to emphasize how important pausing and elongation are to HIV transcription. Additionally, to further support our claim that INTS12KO with AZD5582 & I-BET151 leads to an increase in elongation, that we previously showed with CUT&Tag data showing an increase in total RNAPII seen in within HIV (Figure 5B), we measured RNAPII Ser2 phosphorylation (Figure 5C) and RNAPII Ser5 phosphorylation (Figure 5—figure supplement 2) and added these findings to the manuscript. Upon measuring Ser2 phosphorylation, a marker associated with elongation, we observed evidence of elongation-competent RNAPII in our AZD5582 & I-BET151 condition as well as our INTS12 KO with AZD5582 & I-BET151 condition, as we saw an increase of Ser2 phosphorylation within HIV. Despite seeing elongation-competent RNAPII in both conditions, we only saw a dramatic increase in total RNAPII for our INTS12 KO and AZD5582 & I-BET151 condition (Figure 5B), which supports that there are more elongation events and that an elongation block is overcome specifically with INTS12 KO paired with AZD5582 & I-BET151. This claim is further supported by our data showing an increase in virus in the supernatant only with the INTS12 KO with AZD5582 & I-BET151 condition in cells from PLWH (Figure 6C). We did not observe any statistically significant differences between RNAPII Ser5 phosphorylation, which might be expected as this mark is not associated with elongation (Figure 5—figure supplement 2).

**Reviewer #3 (Public review):**
Summary:Transcriptionally silent HIV-1 genomes integrated into the host`s genome represent the main obstacle to an HIV-1 cure. Therefore, agents aimed at promoting HIV transcription, the so-called latency reactivating agents (LRAs) might represent useful tools to render these hidden proviruses visible to the immune system. The authors successfully identified, through multiple techniques, INTS12, a component of the Integrator complex involved in 3' processing of small nuclear RNAs U1 and U2, as a factor promoting HIV-1 latency and hindering elongation of the HIV RNA transcripts. This factor synergizes with a previously identified combination of LRAs, one of which, AZD5582, has been validated in the macaque model for HIV persistence during therapy (https://pubmed.ncbi.nlm.nih.gov/37783968/). The other compound, I-BET151, is known to synergize with AZD5582, and is a inhibitor of BET, factors counteracting the elongation of RNA transcripts.Strengths:The findings were confirmed through multiple screens and multiple techniques. The authors successfully mapped the identified HIV silencing factor at the HIV promoter.Weaknesses:(1) Initial bias:In the choice of the genes comprised in the library, the authors readdress their previous paper (Hsieh et al.) where it is stated: "To specifically investigate host epigenetic regulators involved in the maintenance of HIV-1 latency, we generated a custom human epigenome specific sgRNA CRISPR library (HuEpi). This library contains sgRNAs targeting epigenome factors such as histones, histone binders (e.g., histone readers and chaperones), histone modifiers (e.g., histone writers and erasers), and general chromatin associated factors (e.g., RNA and DNA modifiers) (Fig 1B and 1C)".From these figure panels, it clearly appears that the genes chosen are all belonging to the indicated pathways. While I have nothing to object to on the pertinence to HIV latency of the pathways selected, the authors should spend some words on the criteria followed to select these pathways. Other pathways involving epigenetic modifications and containing genes not represented in the indicated pathways may have been left apart.(2) Dereplication:From Figure 1 it appears that INTS12 alone reactivates HIV -1 from latency alone without any drug intervention as shown by the MACGeCk score of DMSO-alone controls. If INTS12 knockdown alone shows antilatency effects, why, then were they unable to identify it in their previous article (Hsieh et al., 2023)? The authors should include some words on the comparison of the results using DMSO alone with those of the previous screen that they conducted.(3) Translational potential:In order to propose a protein as a drug target, it is necessary to adhere to the "primum non nocere" principle in medicine. It is therefore fundamental to show the effects of INTS12 knockdown on cell viability/proliferation (and, advisably, T-cell activation). These data are not reported in the manuscript in its current form, and the authors are strongly encouraged to provide them.Finally, as many readers may not be very familiar with the general principles behind CRISPR Cas9 screening techniques, I suggest addressing them in this excellent review: https://pmc.ncbi.nlm.nih.gov/articles/PMC7479249/.

(1) The CRISPR library used was more completely described in a previous publication (Hsieh et al, PLOS Pathogens, 2023). However, we now more explicitly refer the reader to information about the pathways targeted in the library. We also point out how initial hits in the library lead to finding genes outside of the starting library as in the follow-up screen in Figure 7 where each of the members of the INT complex are interrogated even though only INTS12 was the only member in the initial library.

(2) We understand the confusion between the hits in this paper and a previous publication. Indeed, INTS12 was observed in Hsieh et al., PLOS Pathogens, 2023 as a hit in the Venn diagram of Figure 3B of that paper, and in Figure 5A, right panel of that paper. However, it was not followed up on in the previous paper since that paper focused on a hit that was unique to increasing the potency of one particular LRA. We added text to the present manuscript to make it clear that the screens identified many of the same hits. We have also added additional data here on hit validation to underscore the reliability of the CRISPR screen. In one of the cell lines (5A8), EZH2 was a strong hit (Figure 1B). We have now added data that shows that an inhibitor to EZH2 augments the latency reversal of AZD5582/I-BET151 as predicted from the screen. This data has been added to Figure 1, figure supplement 1.

(3) We appreciate the concern that for INTS12 to be a drug target, it should not be essential to cell viability. We now show that knockout of INTS12 has no effect on cell proliferation (new data added in Figure 2—figure supplement 3). In addition, the discussion now adds additional literature references that describe how knockout of INTS12 has relatively minor effects on cell functions in comparison to knockout of other INT members which supports that the proposal that modulation of INTS12 may be more specific than targeting the catalytic modules of Integrator. Nonetheless, we completely agree with the reviewer that many other aspects of how INTS12 affects T cell functions have not been addressed as well as other potential detrimental effect of INTS12 as a drug target in vivo. We now more explicitly describe these caveats in the discussion but feel that the present manuscript is a first step with a long path ahead before the translational potential might be realized.

(4) We now cite the review of CRISPR screens suggested by the reviewer.

**Responses to recommendations for the authors**:
**Reviewer #1 (Recommendations for the authors):**
(1) The authors report in the legend of Figure 2 (and similarly in other figures) that there was "a calculated INTS12 knockout score of 76% (for the one guide used) and 69% (for one of three guides used), respectively." However, it would be helpful to show representative data on the efficiency of INTS12 knockdown in cell lines and primary cells, as well as data on the efficiency of the complementation (Figure 2C).

The knockout scores cited are the genetic assays for the efficiency based on sequence files. As the knockouts are done with multiple guides the knockout for each guide is an underestimate of the total knockout. The complementation, however, was done by adding back INTS12 in a lentiviral vector that also contains a drug resistance marker (puromycin). Cells were then selected for puromycin resistance, and therefore, all of them contain the complemented gene. What one would ideally like is a Western blot to quantify the amount of INTS12 remaining in the knockout pools. Unfortunately, despite obtaining multiple different commercial sources of INTS12 antibodies, we were unable to identify one that was suitable for Western blotting (as opposed to two that did work for CUT&Tag). Nonetheless, the functional data in primary T cells from PLWH and in J-Lat cells lines does show the even if the knockout is suboptimal, we find activation after INTS12 knockout (e.g., Figure 6).

(2) Flow cytometry methods are not reported, but was a viability dye included when testing GFP reactivation (Figure S2)? More broadly, showing data on the viability of cells post-knockdown and drug treatments would help, as cell mortality is inherently associated with latency reactivation in J-Lat cells. For the same reason, reporting viability data would be important for primary cells, as the electroporation procedure can lead to significant mortality.

We did not include viability dyes in the data for GFP activation. However, as described in the public response, we have done growth curves in J-Lat 10.6 cells with and without INTS12 knockout and find no effects on cell proliferation (Figure 2—figure supplement 3). As the reviewer points out, it is not possible to do these experiments in primary cells since the electroporation itself causes a degree of cell death. Nonetheless, we do see effects on HIV activation in these primary cells (Figure 6).

(3) Figure S2 shows a relatively high baseline expression (approximately 15%) of HIV-GFP, which is not unusual for the J-Lat 10.6 clone. However, Figure 3 appears to show no HIV RNA reads in the control condition of this same cell clone. How do the authors reconcile this discrepancy?

We believe that the discrepancies in the flow cytometry versus RNA-seq assays are due to differences in the sensitivity of the assays, the linear range of the assays especially at the lower end, and the different half-lives of RNA versus protein. We now clarify that Figure 3 does not show “no” HIV RNA at baseline, but rather values of ~30 copies per million read counts. This increases to ~800 copies per million read counts when INTS12 knockout cells are treated with AZD5582/I-BET151. These values have the same fold change predicted in Figure 4, and more closely resemble the trend in Figure 2—figure supplement 1.

(4) The combination of AZD5582 and I-BET151 consistently reactivates HIV latency (including GFP protein expression), as previously reported and as shown here by the authors. However, in Figure 5B, RPB3/RNAPII occupancy in the DMSO control appears higher than in the AAVS1KO + AZD5582 and I-BET151 samples. This should be discussed, as it could raise concerns about the robustness of RPB3/RNAPII occupancy results as a proxy for provirus elongation.

As addressed in the public comments, in order to strengthen our claims about transcriptional elongation control, we measured RNAPII Ser2 and Ser5 phosphorylation levels. We see evidence of elongation with Ser2 in the condition of concern (AAVS1 KO + AZD5582 & I-BET151) as well as our main condition of interest (INTS12 KO + AZD5582 & I-BET151) and no change in Ser5 for any condition. With both the Ser2 phosphorylation and total RNAPII as well as our virus release and transcription data we believe that we are seeing evidence of increased elongation with INTS12 KO with AZD5582 & I-BET151. One potential nuance that may not be gathered from the CUT&Tag data is the turnover rate of the polymerase. Despite the levels of RNAPII appearing lower in the condition of concern (AAVS1 KO + AZD5582 & I-BET151) compared to DMSO it is possible that low levels of elongation are occurring but that in our INTS12 KO + AZD5582 & I-BET151 condition there is more rapid elongation and this is why we can observe more RNAPII within HIV. This new data is added in Figure 5C and Figure 5—supplement 2 and its implications are now described in more detail in the discussion.

(5) The authors write that "Degree of reactivation was correlated with reservoir size as donors PH504 (star symbol) and PH543 (upside down triangle) have the largest HIV reservoirs (supplemental Figure S2)." I could not find mention of the reservoir size of these donors in the figure provided.

This confusion was caused by mislabeling of the supplement number, which we fixed, and we added additional labeling to make finding the reservoir size even more clear as this is an important part of the manuscript. This is now found in Supplemental file S4.

**Reviewer #3 (Recommendations for the authors):**
(1) The MAGeCK gene score is a feature that is essential for the interpretation of the results in Figure 1. The authors do quote the Li et al. paper where this score was described for the first time (https://genomebiology.biomedcentral.com/articles/10.1186/s13059-014-0554-4), however, they may understand that not all readers may be familiar with this score. Therefore a didactic short description of this score should be done when introducing the results in Figure 1.

We have added a short description to the paper to address this.

(2) Figure 4. The authors write: "Among the host genes most prominently affected by INTS12 knockout with AZD5582 & I-BET151 are MAFA, MAFB, and ID2 (full list of genes in supplemental file S3)." I am a bit confused. In the linked Excel file there is only a list of a few genes. The differentially expressed genes appear to be many more from Figure 4. The full list should be uploaded.

We believe there was a mistake in our original uploading and naming of the supplements. We have now double-checked numbering on the supplements and added in text clarification of which excel tabs hold the desired information.

(3) Figure 6: The authors are right in highlighting that there is a high level of variability in viral RNA in supernatants in the early stages of viral reactivation. It is therefore advisable to repeat measurements at Day 7, at which variability decreases and data are more reliable (please, see: https://www.thelancet.com/journals/ebiom/article/PIIS2352-3964(23)00443-7/fulltext).

While it would have been nice to prolong these measurements, our current assay conditions are not optimal for longer term growth of the cells. We note that the measurements were all done in biological triplicates (independent knockouts) and in different individuals. Because the number of activatable latent proviruses is variable and the number of cells tested is limiting, the variability in the assays is expected.

(4) Figure 7: The main genes outside the INTS family should be identified, also.

We include the full list in supplemental file S5 and sort by most enriched.

(5) Methods: A statistical paragraph should be added in the Methods section, detailing the data analysis procedures and the key parameters utilized (for example, which is the MAGeCK gene score threshold that they used to consider knockdown efficacy on HIV latency?).

There is no MAGeCK score threshold that we use to determine efficacy on HIV latency. In a previous publication using CRISPR screens for HIV Dependency Factors (Montoya et al, mBio 2023), we showed that there is a relationship between the MAGeCK and the effect of that gene knockout on HIV replication (Figure 5 that paper). However, it is a continuum rather than a strict threshold and we believe that the effects on HIV latency would respond similarly. In the current paper, we have focused on the top hits rather than a comprehensive analysis of all the entire list. In case the reviewer is referring to the average and standard deviation of the non-targeting controls, we have added this to the figure legend and methods.